# SpectrumWorld: Artificial Intelligence Foundation for Spectroscopy

## Abstract

Deep learning holds immense promise for spectroscopy, yet research and evaluation in this emerging field often lack standardized formulations. To address this issue, we introduce SpectrumLab, a pioneering unified platform designed to systematize and accelerate deep learning research in spectroscopy. SpectrumLab integrates three core components: a comprehensive Python library featuring essential data processing and evaluation tools, along with leaderboards; an innovative SpectrumAnnotator module that generates high-quality benchmarks from limited seed data; and SpectrumBench, a multi-layered benchmark suite covering 14 spectroscopic tasks and over 10 spectrum types, featuring spectra curated from over 1.2 million distinct chemical substances. Thorough empirical studies on SpectrumBench with 23 cutting-edge multimodal LLMs reveal critical limitations of current approaches. We hope SpectrumLab will serve as a crucial foundation for future advancements in deep learning-driven spectroscopy. The anonymous code and experimental records are available at `https://anonymous.4open.science/r/SpectrumLab-8C4E/`.

## 1 Introduction

Spectroscopy, which investigates the interaction between electromagnetic radiation and matter, provides a powerful way to investigate the molecular structure and properties (Elias et al., 2004; Prasad et al., 2025). By capturing characteristic patterns, such as peaks and shifts, in signals analogous to audio waveforms, spectroscopy offers a compact, information-rich representation of molecular systems (Ralbovsky & Lednev, 2020). This low-dimensional encoding is indispensable in chemistry (Silber et al., 2016; Seo et al., 2017), and life sciences (Ralbovsky & Lednev, 2020; Zhang et al., 2023; Gasparin et al., 2025). It is not only central to molecular structure elucidation (*i.e.*, Spectrum-to-Molecule structure) and property prediction, but also a key enabler for new material discovery and drug screening. In recent years, machine learning methods, especially deep learning, have demonstrated tremendous potential in spectroscopic data analysis, opening a new era of automation and intelligence in spectroscopy research (Gastegger et al., 2017b; Gerrard et al., 2019; Fine et al., 2020; Han et al., 2022; Zou et al., 2023b; Devata et al., 2024; Lu et al., 2025).

Despite recent advances, deep learning for spectroscopy still faces several fundamental challenges. Specifically, high-quality experimental spectral data remain scarce and expensive to acquire (van de Sande et al., 2023; Flanagan et al., 2025), leading to public datasets that are limited in size and suffer from highly imbalanced distributions (Bongiorno et al., 2022; Stenning et al., 2024; Peng et al., 2025), which severely restricts model generalization. In addition, a substantial domain gap exists between experimental and computational spectra due to complex measurement conditions (Agarwala et al., 2022), hindering the deployment of models trained on theoretical data. Furthermore, spectroscopy is inherently multimodal: it encompasses various spectral types (*e.g.*, infrared, Raman, nuclear magnetic resonance) represented as either 1D signals or 2D images, often requiring integration with other molecular modalities such as molecular graphs, SMILES strings, and 3D conformations (Litsa et al., 2021; Devata et al., 2024). The heterogeneous nature and semantics of these data modalities pose significant challenges for deep learning systems. Finally, the field lacks standardized benchmarks, with a fragmented landscape of tasks and datasets making it difficult to systematically evaluate and compare model performance.

To address these challenges, we introduce SpectrumLab, a modular platform that streamlines the entire lifecycle of AI-driven spectroscopy from data preprocessing to model evaluation. Built on top of SpectrumLab, we construct SpectrumBench, a unified benchmark suite designed to evaluate machine learning models across diverse spectroscopic tasks and modalities. In contrast to existing approaches such as DiffSpectra (Wang et al., 2025b) and MolSpectra (Wang et al., 2025a), which rely on contrastive learning and diffusion architectures, we are among the first to incorporate multi-modal large language models (MLLMs) into spectroscopic learning, using their alignment capabilities to bridge heterogeneous data modalities.

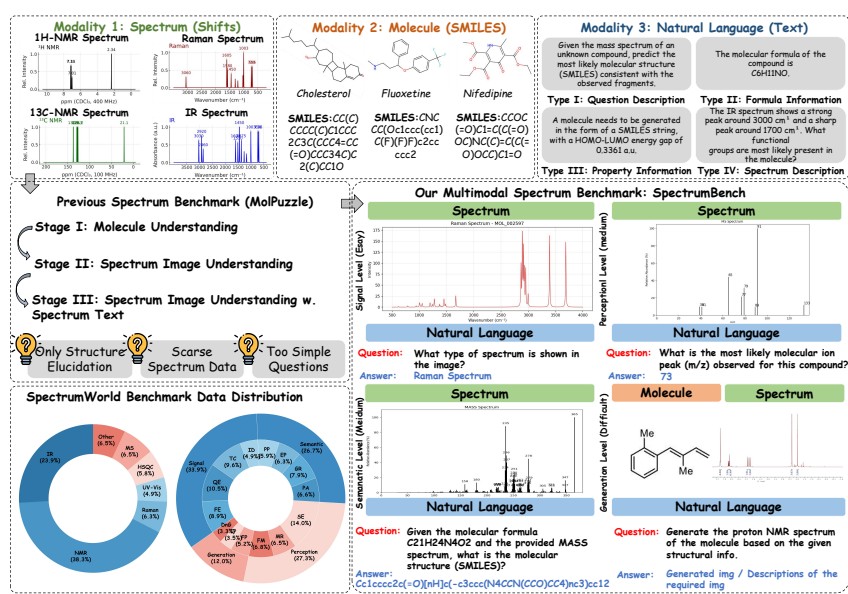

Figure 1: Overview of SpectrumBench.

Our main contributions are: (1) We introduce **SpectrumLab**, the first standardized framework tailored for spectroscopic machine learning with multimodal large language models, enabling reproducible pipelines from raw spectra to evaluation. (2) We design **SpectrumAnnotator**, an automatic benchmark generator that constructs task-specific datasets from spectrum seeds, greatly accelerating prototyping and stress-testing of new models. (3) We release **SpectrumBench**, a large-scale benchmark suite covering diverse spectroscopic modalities and tasks, accompanied by unified evaluation protocols and public leaderboards to foster fair comparison and community progress.

## 2 RELATED WORK

Figure 2: Representative SpectraML methods categorized by **Spectral Type (left Y-axis)** and **Model Type (right Y-axis)**. Each dot indicates the use of a specific spectral modality or model architecture in a given method. Note that Raman is not included; thus, methods using it (*e.g.*, DeepCID (Fan et al., 2019)) are not shown on the left Y-axis.

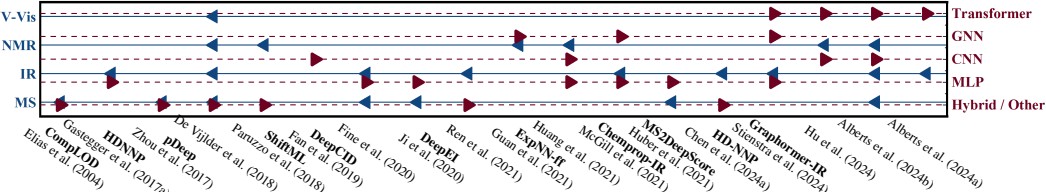

**Machine Learning for Spectroscopy**. Spectroscopy is fundamental for molecular structure analysis and scientific discovery, enabling insights into chemical properties and interactions (Guo et al., 2025). Its applications span diverse scientific domains, including chemistry, material science, and drug development (Shao et al., 2025; Sun et al., 2025). Machine learning techniques have been

extensively applied in spectroscopy for tasks such as molecular structure elucidation (spectrum-to-molecule) (Kuhn et al., 2008; De Vijlder et al., 2018; Paruzzo et al., 2018; Fan et al., 2019; Nguyen et al., 2019; Ji et al., 2020; Fine et al., 2020; Huang et al., 2021; Hu et al., 2024; Lu et al., 2025; Liu et al., 2025) and spectral simulation (molecule-to-spectral) (Gastegger et al., 2017a; Zhou et al., 2017; Liu et al., 2017; McGill et al., 2021; Guan et al., 2021; Ren et al., 2021; Young et al., 2024). As illustrated in Figure 2, recent efforts have explored a variety of spectral modalities, such as IR (Hu et al., 2024), NMR (Hu et al., 2024), UV-Vis (De Vijlder et al., 2018), MS (Huber et al., 2021), and Raman (Fan et al., 2019), and have adopted heterogeneous deep learning model architectures, ranging from MLPs (Stienstra et al., 2024) and CNNs (Alberts et al., 2024b) to GNNs (McGill et al., 2021) and Transformers (Alberts et al., 2024a). Despite these rapid progresses, existing methods still face several limitations: (1) most studies are constrained to a single modality (*e.g.*, IR or MS), lacking generalization across spectral types (Beck et al., 2024); (2) the field lacks unified benchmarks and evaluation protocols, making objective comparisons difficult; (3) dataset sizes remain limited and imbalanced, further impeding reproducibility and robustness; (4) previous benchmarks does not support multi-modal large language models. These limitations highlight the need for standardized, cross-modal frameworks to advance machine learning for spectroscopy, especially spectroscopy foundation models.

**Spectroscopy Foundation Models**. While foundation models have shown promising progress in scientific discovery (Tan et al., 2025; Xia et al., 2025), spectroscopy foundation models are still underexplored. This is largely due to the inherent multimodal nature of spectroscopic data, which combines spectral signals with diverse molecular representations. Although recent efforts such as SpectraFM (Koblischke & Bovy, 2024) and LSM1-MS2 (Asher et al., 2024) have introduced pre-trained foundation models on Stellar and MS spectra for chemical property prediction, these models remain fundamentally single-modal, focusing solely on spectral information. Despite these challenges, the integration of spectroscopy into the foundation model paradigm holds significant promise for advancing automated analysis and multi-modal scientific discovery in the future.

Table 1: Comparison of Benchmark Studies. **Notes:** "Other" in the Spectral Modality column includes modalities not explicitly listed, such as HSQC (Heteronuclear single quantum coherence spectroscopy) and UV-Vis (Ultraviolet-visible spectroscopy). The NMR column refers to both $^1$H-NMR and $^{13}$C-NMR. We unify tasks' terminology for clarity.

| Benchmark | Reference | Spectral Modality | | | | | Task | | | Understanding | | | |
|---|---|---|---|---|---|---|---|---|---|---|---|---|---|
| | | Raman | IR | NMR | MS | Other | Molecular Elucidation | Spectrum Simulation | *De novo* Generation | GR | PA | FM | MR |
| NovoBench | (Zhou et al., 2024a) | | | | ✓ | | | | ✓ | | | | |
| MolPuzzle | (Guo et al., 2024b) | | ✓ | ✓ | ✓ | | ✓ | | | ✓ | | | |
| Multimodal Spec | (Alberts et al., 2024b) | | ✓ | ✓ | ✓ | ✓ | ✓ | ✓ | | ✓ | | | |
| MassSpecGym | (Bushuiev et al., 2024b) | | | | ✓ | | ✓ | ✓ | | | | | |
| NMRNet | (Xu et al., 2025) | | | ✓ | | | | | | | ✓ | | |
| ViBench | (Lu et al., 2025) | ✓ | ✓ | | | | ✓ | | | | | | |
| SpectrumBench | Ours | ✓ | ✓ | ✓ | ✓ | ✓ | ✓ | ✓ | ✓ | ✓ | ✓ | ✓ | ✓ |

**Abbreviations:** GR = Functional Group Recognition, PA = Peak Assignment, FM = Fusing Spectroscopic Modalities, MR = Multimodal Molecular Reasoning.

**Benchmark and Toolkits for Spectroscopy**. Several benchmarks and toolkits have been developed to support spectroscopic machine learning research (Heid et al., 2023; Zhou et al., 2024b; Bushuiev et al., 2024a; Guo et al., 2024b; Devata et al., 2024; Ruan et al., 2024; Guo et al., 2025). However, many of these efforts remain limited in scope (either spectrum modalities or tasks), lacking extensibility and comprehensive evaluation across diverse spectroscopic tasks and modalities. For example, MassSpecGym (Bushuiev et al., 2024a) focuses solely on MS data and does not incorporate language descriptions, hindering support for multi-modal inputs. Although MolPuzzle (Guo et al., 2024b) enables multi-modal inputs, it omits Raman spectra and lacks support for pure spectral understanding tasks. Furthermore, several toolkits (Bushuiev et al., 2024a; Zhou et al., 2024b) do not provide interfaces for multi-modal large language models (MLLMs), and even MolPuzzle lacks benchmarking for more recent MLLMs. In contrast, our SpectrumLab is a unified, extensible, and reproducible platform that addresses these limitations by supporting a wide range of spectroscopic tasks, modalities, and integration with MLLMs. Table 1 systematically compares representative studies in terms of their spectral modality and task coverage. SpectrumLab not only fills critical gaps in data, evaluation, and tooling, but also establishes a new standard for spectroscopic AI and enables future advances in multi-modal, large-model-driven scientific discovery.

## 3 SPECTRUMBENCH

**Overview.** SpectrumBench is a unified benchmark suite for deep learning in spectroscopy, covering four hierarchical levels and 14 sub-tasks that span from spectroscopy understanding to generation. All questions and tasks are initially defined by domain experts, and subsequently refined and validated through expert review and rigorous quality assurance processes. Compared to existing benchmarks, SpectrumBench offers broad modality and task coverage within a standardized, extensible framework for fair and reproducible model evaluation.

**Spectroscopic Type.** Unlike previous benchmarks that are limited to a single spectroscopic modality or narrowly defined data types (Bushuiev et al., 2024b), SpectrumBench integrates a diverse array of spectroscopic data sources. Our SpectrumBench benchmark currently includes more than 10 distinct types of spectroscopic data, such as infrared (IR), nuclear magnetic resonance (NMR), and mass spectrometry (MS). As illustrated in Figure 1, this comprehensive data foundation accurately reflects the diverse and complex multi-modal spectroscopic scenarios encountered in real-world applications.

Table 2: Tasks' categories and statistics.

| Category | Task | # questions |
|---|---|---|
| Signal | Spectrum Type Classification (TC) | 55 |
| | Spectrum Quality Assessment (QE) | 60 |
| | Basic Feature Extraction (FE) | 51 |
| | Impurity Peak Detection (ID) | 28 |
| Perception | Functional Group Recognition (FG) | 45 |
| | Elemental Compositional Prediction (EP) | 36 |
| | Peak Assignment (PA) | 38 |
| | Basic Property Prediction (PP) | 34 |
| Semantic | Molecular Structure Elucidation (SE) | 80 |
| | Fusing Spectroscopic Modalities (FM) | 39 |
| | Multimodal Molecular Reasoning (MR) | 37 |
| Generation | Forward Problems (FP) | 30 |
| | Inverse Problems (IP) | 20 |
| | *De Novo* Generation (DnG) | 19 |

**Task.** In contrast to previous benchmarks that primarily focus on molecule elucidation or spectrum simulation, SpectrumBench encompasses a much broader spectrum of task types. SpectrumBench is organized according to a multi-level hierarchical taxonomy that systematically covers tasks ranging from low-level signal analysis to high-level semantic reasoning and generative challenges. This taxonomy, developed through expert consultation and iterative refinement, comprises four principal layers: **signal, perception, semantic, and generation**. Each layer is further divided into several subcategories, capturing a diverse set of scientific and application-driven tasks. Detailed definitions and representative examples for each task layer are provided in the Appendix C.

### 3.1 DATA CURATION PIPELINE

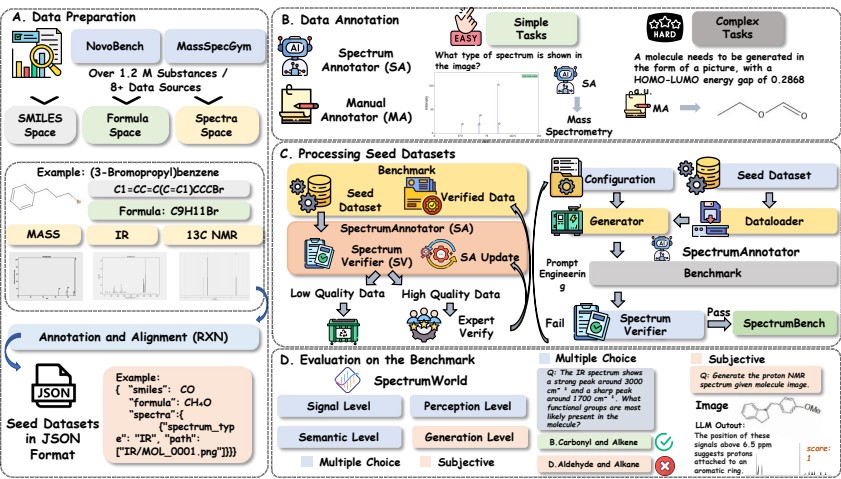

Figure 3: Overview of the data curation pipeline used in SpectrumBench.

**Task Construction.** Spectroscopic machine learning encompasses a wide spectrum of tasks, driven by the intrinsic complexity of molecular structures and the multifaceted nature of spectroscopic data. These tasks often involve diverse input modalities (e.g., molecular graphs, SMILES, textual prompts) and equally varied outputs (e.g., spectra, chemical attributes, structured predictions), which

reflect the real-world demands of chemical analysis, property reasoning, and molecular generation. To illustrate this diversity, we organize existing spectroscopic tasks into four broad input-output categories: (1) *Molecule-to-Spectrum (Spectrum Simulation)* aims at generating a spectrum based on molecular structure. (2) *Spectrum-to-Molecule (Molecule Elucidation)* refers to the tasks that infer molecular structures from spectra. (3) *Text-to-Any[1] (De novo Generation)* refers to the task of generating novel, diverse, and reasonable molecular structures (SMILES string, 2D molecular graph) and/or predicting multimodal information (spectra, properties) according to specific goals (*e.g.*, molecules of a specific nature, ligands of a specific target). Moreover, in previous studies, many tasks involving inferring molecular structures from spectra were also categorized under "*de novo* generation" (Bushuiev et al., 2024a; Lu et al., 2025). While this has some rationality, for the sake of consistency in our task framework, we clarify that our defined *de novo* generation task has distinct characteristics: its input consists solely of textual descriptions, which may include specifications of molecular properties (*e.g.*, desired chemical natures, target-binding affinities), without involving spectral data as input. Meanwhile, the output scope is broader, encompassing not only molecular structures but also spectral and textual descriptions of molecules. (4) *Any-to-Text (Understanding)*. Tasks in signal, perception, semantic layers fall under the "Understanding" category. Its task type is presented in the form of multiple-choice questions, which may include tasks such as inferring the molecular structure from spectrum (*e.g.*, functional group recognition, peak attribution tasks). This partially overlaps with the molecular elucidation tasks described above. For a compromise design, we use the output form to distinguish between them. The question format of "Understanding" tasks will only be multiple-choice questions, which means the output is text.

**Taxonomy Definition**. These input-output patterns offer a high-level overview of the task landscape. However, previous works often cover only a subset of them, limiting both their generalizability and their ability to benchmark diverse ML capabilities. We show these patterns in Table 1, which highlights substantial heterogeneity between existing methods. To address this limitation and support more structured and extensible benchmarking, we propose a four-level hierarchical taxonomy tailored to spectroscopic machine learning: *Signal*, *Perception*, *Semantic*, and *Generation*—is designed to reflect the logic of real-world scientific workflows in spectroscopy. As depicted in Figure 3, this layered structure systematically provides a robust framework for our 14 meticulously designed tasks detailed in Table 2.

(1) *Signal level:* This foundational layer focuses on the direct analysis and processing of raw spectral data, such as spectrum type classification and peak detection. Tasks at this level are designed to extract and refine primary features from experimental measurements, mirroring the initial steps taken by chemists to prepare and interpret spectra in the laboratory. This level primarily encompasses Any-to-Text(Understanding) tasks that operate directly on raw signal data.

(2) *Perception level:* Building upon the processed signals, the perception layer addresses pattern recognition and intermediate interpretation tasks, such as functional group identification, peak assignment, and basic molecule properties prediction. This stage reflects the chemist's effort to translate spectral features into meaningful chemical information, bridging the gap between raw data and higher-level understanding. Many Any-to-Text(Understanding) tasks that involve interpreting specific patterns within spectra fall into this category.

(3) *Semantic level:* At this layer, the focus shifts to comprehensive molecular reasoning and property inference, including molecule elucidation and cross-modal correlation (*e.g.*, linking spectra to molecular graphs or textual descriptions). The semantic layer encapsulates the core scientific reasoning that underpins hypothesis generation and validation in spectroscopic research, primarily addressing advanced Any-to-Text(Understanding) tasks that require intricate chemical knowledge and contextualization.

(4) *Generation level:* The final layer encompasses creative and generative tasks, where new entities are produced. The level explicitly consolidates all tasks involving the synthesis of new data or structures, including Molecule-to-Spectrum (*e.g.*, spectrum generation from molecular inputs), Spectrum-to-Molecule (*e.g.*, generates a molecular structure from spectra input). These tasks emulate advanced scientific workflows where new hypotheses, molecules, or spectral data are generated to drive discovery and innovation.

---

[1]"Any" encompasses various data modalities, such as molecular representations, spectral data, or peptide sequences.

Figure 4: SpectrumLab is the main component of SpectrumWorld, which could be extended conveniently. More multi-modal spectrum data could be included, as well as MLLMs.

**Seed Data Preparation**. The seed datasets for this work were curated from three primary sources: proprietary data, public repositories, and literature mining. As illustrated in Figure 3, the construction of seed datasets begins with the aggregation of raw data from multiple sources. All collected datasets undergo a unified processing pipeline that systematically maps each entry into three core chemical spaces: SMILES string, molecular formula, and spectra. Rigorous cleaning, normalization, and deduplication ensure data consistency and reliability. Following this, human annotation and alignment are performed to guarantee scientific accuracy and completeness. The resulting seed datasets are organized at the level of individual chemical substances, with each record containing the compound's SMILES, molecular formula, and a structured set of associated spectra, all stored in a standardized JSON format to facilitate downstream annotation. Detailed descriptions of the seed datasets and the standardization process are provided in Appendix I. We ensure that the curated seed data are not contaminated.

**Data Annotation**. We use two annotation methods: automated and manual annotation. *(1) Automated annotation (SpectrumAnnotator)*. For tasks characterized by well-defined rules like spectrum recognition , we design SpectrumAnnotator, a core contribution of this work. SpectrumAnnotator is a novel, self-developed annotation framework that harnesses the zero-shot and multi-modal reasoning capabilities of state-of-the-art MLLMs. Given curated seed datasets and a set of pre-defined benchmark prompts, SpectrumAnnotator automatically designs and generates high-quality, multi-modal benchmark data. Further technical details and implementation specifics of SpectrumAnnotator are provided in Appendix D. *(2) Manual Annotation*. For more complex or open-ended tasks, particularly those involving multi-step reasoning or sophisticated scientific interpretation, manual annotation by domain experts is indispensable. Human annotators ensure the scientific validity and depth of the benchmark, especially in cases where automated methods cannot handle.

**Data Quality Assurance**. To ensure the integrity and reliability of SpectrumBench, we implement a comprehensive quality assurance pipeline, as illustrated in Figure 3. The process begins with *Candidate Data* undergoing automated screening by the *SpectrumVerifier* (**SV**). This stage efficiently detects and filters out clear errors such as missing options or image-text discrepancies, categorizing them as *Low Quality Data* for removal. Remaining *High Quality Data* proceeds to expert manual evaluation. If issues are identified, a feedback loop through internal annotator update initiates targeted reannotation via *SpectrumAnnotator* (**SA**). This multi-stage quality control ensures only high-quality, scientifically robust data are included in our final benchmark.

## 4 SPECTRUMLAB

### 4.1 SYSTEM OVERVIEW

**AI-ready Datasets and AI-solvable Tasks**. SpectrumLab is tightly integrated with SpectrumAnnotator, which is responsible for generating high-quality benchmarks from seed datasets collected from diverse sources. In this workflow, SpectrumAnnotator curates a wide range of scientifically rigorous benchmarks from these seeds. SpectrumLab then offers a flexible abstraction for users to define and encapsulate specific AI-solvable tasks based on these curated benchmarks. A core abstraction unique to SpectrumLab is the *Benchmark Group*. Users can combine multiple benchmark

instances or select specific subsets to form a *Benchmark Group*, creating tailored task definitions within a unified framework. By encapsulating benchmarks as tasks, SpectrumLab streamlines the process of task definition and evaluation.

**Toolkits and Ecosystem**. SpectrumLab offers a flexible ecosystem of Python libraries and tools designed to streamline the entire workflow for spectroscopy, from data preprocessing to model evaluation. Its modular design allows seamless integration of custom models and tasks. Distributed via the Python Package Index(PyPI) for easy installation, SpectrumLab provides a comprehensive environment for state-of-the-art machine learning research in spectroscopy.

**Leaderboards**. To ensure transparency and reproducibility, SpectrumLab incorporates a comprehensive public leaderboard system that systematically tracks and compares the performance of a wide range of models across all tasks. The leaderboard provides fine-grained reporting, recording each model's results on both high-level and detailed tasks. The platform currently supports benchmarking for over 20 MLLMs, including prominent open-source models such as InternVL3 (Zhu et al., 2025) and proprietary models like GPT-4o (OpenAI, 2024), across 14 specific tasks.

## 4.2 MODULAR DESIGN

| Model | Signal | | | | Perception | | | | Semantic | | | Generation | | | Avg. |
|---|---|---|---|---|---|---|---|---|---|---|---|---|---|---|---|
| | TC | QE | FE | ID | GR | EP | PA | PP | SE | FM | MR | FP | IP | DnG | Perf |
| *Closed-source MLLMs* | | | | | | | | | | | | | | | |
| Claude-3.5-Sonnet | 96.36 | 28.33 | 76.47 | 71.43 | 60.00 | 77.78 | 76.32 | 85.29 | 82.50 | 69.23 | 94.59 | 20.00 | 0 | 0 | 59.88 |
| Claude-3.7-Sonnet | 96.36 | **38.33** | 86.27 | 82.14 | 71.43 | **88.89** | 71.05 | 88.24 | 82.28 | 74.36 | 89.19 | 20.00 | 0 | 5.26 | 63.84 |
| Claude-4-Sonnet | 96.36 | 35.00 | 88.24 | 92.86 | 62.22 | 63.89 | 60.53 | 76.47 | 16.25 | 43.59 | 64.86 | 3.33 | 0 | 21.05 | 51.76 |
| Claude-3.5-Haiku | 94.55 | 31.67 | 50.98 | 92.86 | 66.67 | 75.00 | 76.32 | 76.47 | 67.50 | 64.10 | 81.08 | 10.00 | 0 | 0 | 56.23 |
| Claude-4-Opus | 96.36 | 33.33 | 86.27 | 92.86 | 73.33 | 83.33 | 71.05 | 85.29 | 32.50 | 76.92 | 86.49 | 16.67 | 0 | 5.26 | 59.98 |
| GPT-4o | 96.36 | 33.33 | 68.63 | 92.86 | 59.88 | 77.78 | 63.16 | 79.41 | 78.75 | 58.97 | 89.19 | 10.00 | 0 | 0 | 57.74 |
| GPT-4.1 | 94.55 | 28.33 | 86.27 | 85.71 | 53.33 | 77.78 | 63.16 | 79.41 | 82.50 | 66.67 | 91.89 | 33.33 | 10.53 | 0 | 60.96 |
| GPT-4-Vision | 94.55 | 33.33 | 72.55 | 92.86 | 73.33 | 72.22 | 71.05 | 82.35 | 73.75 | 53.85 | 97.30 | 23.33 | 5.00 | 0 | 60.39 |
| Gemini-2.5-pro | 96.36 | 35.00 | 90.20 | 67.86 | **75.56** | 86.11 | 65.79 | 79.41 | 68.75 | **84.62** | 97.30 | 50.00 | 5.00 | **47.37** | 67.81 |
| Grok-2-Vision | 94.55 | 31.67 | 74.51 | 89.29 | 64.44 | 80.56 | 73.68 | 82.35 | 37.50 | 66.67 | 81.08 | 23.33 | 0 | 0 | 57.12 |
| Qwen-VL-Max | 94.55 | 36.67 | 90.20 | 92.86 | 60.00 | 80.56 | 78.95 | 88.24 | 32.50 | 71.79 | 91.89 | 43.33 | 0 | 5.26 | 61.91 |
| Doubao-1.5-Vision-Pro | 98.18 | 33.33 | 78.43 | 92.86 | 66.67 | 83.33 | 68.42 | 88.24 | 67.50 | 56.41 | 81.08 | 6.67 | 0 | 0 | 59.23 |
| Doubao-1.5-Vision-Pro-Thinking | 96.36 | 35.00 | 78.43 | 67.86 | 53.33 | 80.56 | 73.68 | **91.18** | 68.75 | 66.67 | 91.89 | **66.67** | 5.00 | 5.26 | 62.90 |
| *Open-source MLLMs* | | | | | | | | | | | | | | | |
| Qwen2.5-VL-32B-Instruct | 92.73 | 26.67 | 37.25 | 71.43 | 57.78 | 44.44 | 31.58 | 61.76 | 0.00 | 5.13 | 45.95 | 20.00 | 0 | 0 | 35.34 |
| Qwen2.5-VL-72B-Instruct | 94.55 | **38.33** | 86.27 | 92.86 | 42.22 | 80.56 | 78.95 | 88.24 | 66.25 | 76.92 | 91.89 | 30.00 | 0 | 10.53 | 62.68 |
| InternVL3-78B | 96.36 | **38.33** | 70.59 | 71.43 | 48.49 | 75.00 | **81.58** | 88.24 | 62.50 | 69.23 | 83.78 | 23.33 | 0 | 5.26 | 58.15 |
| InternVL3.5-241B | 98.18 | 33.33 | 90.20 | 92.86 | 66.67 | 77.78 | 71.05 | 85.29 | 86.25 | 61.54 | **100.00** | 33.33 | 10.00 | 10.53 | 65.50 |
| Llama-3.2-11B-Vision-Instruct | 34.55 | 11.67 | 13.73 | 25.00 | 20.00 | 41.67 | 15.79 | 29.41 | 7.50 | 5.13 | 21.62 | 0 | 0 | 0 | 16.15 |
| Llama-3.2-90B-Vision-Instruct | 38.18 | 10.00 | 35.29 | 25.00 | 17.78 | 27.78 | 28.95 | 20.59 | 21.25 | 5.13 | 43.24 | 0 | 0 | 0 | 19.51 |
| DeepSeek-VL2 | 52.73 | 23.33 | 29.41 | 28.57 | 8.89 | 27.78 | 28.95 | 50.00 | 15.00 | 15.38 | 32.43 | 10.00 | 5.00 | 5.26 | 23.77 |
| GLM-4.5V | **100.00** | 28.33 | 70.59 | 92.86 | 73.33 | 83.33 | 71.05 | **91.18** | 63.75 | 69.23 | 83.78 | 0 | 0 | 10.53 | 59.85 |
| InternS1-nothink | 98.18 | 36.67 | 72.55 | 89.29 | 51.11 | 72.22 | 73.68 | 79.41 | 86.25 | 66.67 | 94.59 | 13.33 | 0 | 0 | 59.57 |
| InternS1-think | 98.18 | 25.00 | 80.39 | 89.29 | 64.44 | **88.89** | 73.68 | **91.18** | **90.00** | 56.41 | 91.89 | 10.00 | **40.00** | 15.79 | 65.37 |
| **Overall Avg.** | 89.09 | 30.65 | 70.16 | 77.95 | 56.13 | 71.62 | 63.84 | 76.85 | 56.08 | 55.85 | 79.79 | 20.29 | 3.50 | 6.41 | 54.15 |

Table 3: Accuracies (%, ↑) of all models on different levels. Task abbreviations (e.g., TC, QE, FE, etc.) are defined in Table 2. **best: bold**, second best: underlined. The second last column calculates the arithmetic mean and the last column calculates the true weighted mean of each row.

SpectrumLab adopts a modular architecture to maximize flexibility and extensibility. The core components include:

(1) **Benchmark Group:** SpectrumLab organizes SpectrumBench into hierarchical groups corresponding to different levels of spectroscopic reasoning. This structure enables systematic evaluation across various tasks and spectroscopic modalities, while also supporting rapid assessment of specialized models on domain-specific spectroscopic tasks.

(2) **Model Integration:** SpectrumLab offers a unified, extensible framework for integrating external models. Using standardized APIs and modular adapters, it connects seamlessly to a wide range of model types, from cloud-based services to locally deployed solutions, enabling consistent benchmarking within a single evaluation environment.

(3) **Evaluator:** Serving as the abstract core of the benchmark evaluation engine, the Evaluator module in SpectrumLab is designed for flexible and extensible assessment of model performance across diverse spectroscopic tasks. It enables the customization of evaluation metrics and protocols according to the specific requirements of each task, and can be seamlessly integrated with both the *Benchmark Group* and external model modules. This modular abstraction allows researchers to define and implement tailored evaluation strategies, ensuring rigorous and task-appropriate benchmarking.

Currently, SpectrumLab supports the following two types of evaluators: (i) *Choice Evaluator:* Specially designed for multiple-choice tasks. (ii) *Open Evaluator:* Targeted at generative tasks, this evaluator supports flexible assessment protocols, enabling comprehensive evaluation of free-form and creative model outputs.

## 5 EXPERIMENT

### 5.1 BENCHMARK SETUP

For signal-, perception-, and semantic-level tasks, SpectrumBench standardizes them into a multiple-choice question format, with each question having four options. A correct answer is scored as 1, and an incorrect answer is scored as 0. Generation-level tasks usually do not have fixed-form answers. For Molecule-to-Spectrum tasks, the input is a molecule, and the output is a spectrum. For Spectrum-to-Molecule tasks, the input consists of multiple spectral images, and the output is a molecule. We aim to encourage models to generate meaningful reasoning trajectories rather than simply providing a final answer. This approach can help circumvent the issue of data leakage. Therefore, we use an additional MLLM to score the responses following these steps: (1) Model predictions that do not conform to the specified output format for a given question are assigned a score of zero. (2) For predictions meeting the required format, a dedicated scoring model evaluates the model's output against the answer, assigning a score normalized between 0 and 1. GPT-4o is employed as the scoring model in our experiment. This design standardizes the primary evaluation metric across all tasks in SpectrumBench to accuracy (%). Leveraging SpectrumLab's flexible model interface, we integrated 23 leading open- and closed-source MLLMs for our experiments. Further details on benchmarking candidates and cost analysis are provided in Appendix F and J, respectively.

### 5.2 MAIN FINDINGS

We draw several key insights from the results in Table 3.

(1) **Task complexity reveals model capabilities and limitations.** Models exhibit strong foundational capabilities in basic tasks, with Signal and Perception tasks showing robust performance across all models. Spectrum Type Classification(TC) achieves an average accuracy of 89.09%, while Impurity Peak Detection (ID) shows an average of 77.95%. However, performance significantly declines in more complex tasks, particularly within the Generation category, which shows an average accuracy of only 6.41%. Within the Generation level, there are notable performance differences: FP achieves an average of 20.29%, significantly outperforming Inverse Problems (IP) at 3.50% and *De Novo* Generation (DnG) at 6.41%. This suggests that models are more adept at forward prediction tasks (molecule-to-spectrum) than inverse problems. QE tasks prove particularly challenging, with an average of 30.65% across all models, and many models scoring 0% in IP and DnG tasks. This performance pattern reveals a clear hierarchy: models excel at basic pattern recognition and signal processing but struggle with advanced reasoning, creative generation, and complex cross-modal synthesis tasks that require deeper scientific understanding.

(2) **Closed-source models lead overall performance with gemini-2.5-pro achieving best results.** Gemini-2.5-pro emerges as the top-performing model with an overall average accuracy of 67.81%, securing top-2 scores in 6 out of 14 tasks. The model demonstrates exceptional capabilities across multiple dimensions: it leads in Functional Group Recognition (GR) with 75.56%, and ranks second in several other tasks, including Elemental Compositional Prediction(EP) and Forward Problems(FP). Closed-source models generally maintain a performance advantage. However, this gap is narrowing, models like InternVL3.5-241B(65.50%) and InternS1-think (65.37%) are approaching or even surpassing some closed-source counterparts.

(3) **Reasoning capabilities drive generation task performance.** Doubao-1.5-Vision-Pro-Thinking demonstrates exceptional performance in generation tasks, achieving 66.67% accuracy in Forward Problems (FP), significantly outperforming the second-best closed-source model (Gemini-2.5-pro at 50.00%). This remarkable 16.67% point advantage highlights the critical role of advanced reasoning capabilities in complex molecule generation tasks. InternS1-think also outperforms InternS1 (65.37% vs. 59.57%). This superior performance suggests that the "thinking" mode is essential for tackling sophisticated cross-modal scientific reasoning challenges.

## 5.3 ERROR ANALYSIS

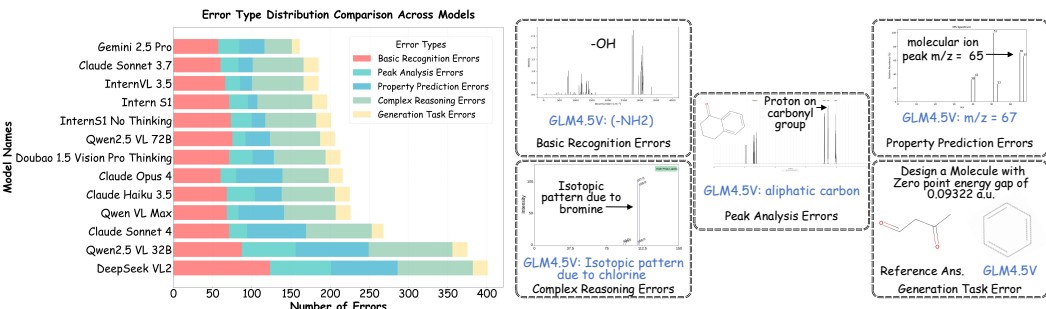

Figure 5: Error types and distributions in SpectrumBench.

**Error Analysis.** For our error analysis, we group the 14 tasks in *SpectrumBench* into five families: *Basic recognition* (elemental composition, functional-group recognition, spectrum-type classification, spectrum-quality assessment), *Peak analysis* (peak assignment, impurity-peak detection, basic feature extraction), *Property prediction* (basic property prediction, molecular structure elucidation), *Complex reasoning* (multimodal molecular reasoning, fusing spectroscopic modalities, forward/inverse problems), and *Generation* (de novo spectrum generation).

**Family-wise minima.** On **Basic recognition**, **Gemini-2.5-Pro** attains the lowest error (**29.1%**), indicating comparatively stronger grounding in spectrum type/quality and functional-group cues. For **Peak analysis**, **Qwen2.5-VL-72B** achieves the lowest error (**14.5%**), suggesting effective handling of isotopic/fragment patterns and impurity peaks. Within **Property prediction**, **Intern-S1** yields the best result (**10.5%**), followed by **InternVL-3.5** (**13.2%**); both exhibit more reliable mapping from spectral evidence to molecular properties/structures. The **Complex reasoning** slice is the sub-most challenging: although **Gemini-2.5-Pro** leads with **30.7%** error, the majority of models exceed 50% in this family, underscoring difficulties with long-horizon, cross-modal deduction. For **Generation**, **Gemini-2.5-Pro** again performs best (**52.6%** error), while several models approach failure on nearly all instances (errors near 100%).

**Observations and implications.** The error profiles reveal two principal bottlenecks: (i) *low-level spectral grounding* (spectrum type/quality and functional-group perception) and (ii) *multi-step symbolic integration* across modalities and tasks. The former dominates early-stage perception failures that cascade to peak interpretation, whereas the latter manifests as brittle chains when executing forward/inverse reasoning or modality fusion. We hypothesize that tighter coupling to spectroscopic priors (fragmentation and isotopic rules, impurity models) and reasoning-aware supervision (tool-augmented peak→property mappings, intermediate targets) are necessary to reduce both recognition errors and brittle deduction.

## 6 CONCLUSION

In this work, we have presented two key contributions to advance machine learning in spectroscopy: SpectrumBench and SpectrumLab. SpectrumBench is a comprehensive, extensible benchmark suite covering over 10 spectrum modalities and 14 tasks, grounded in real-world chemical practices, enabling rigorous and reproducible evaluation across hierarchical taxonomy (signal, perception, semantic, generation). SpectrumLab is a unified, modular platform for dataset management, annotation, evaluation, and public leaderboards, offering a robust Python ecosystem with standardized interfaces that significantly lower the barrier for developing and deploying advanced models. Together, SpectrumBench and SpectrumLab set a new standard for spectroscopic machine learning, fostering systematic comparison, reproducibility, and innovation, and catalyzing future research for more powerful and interpretable models.

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

# A ABLATION STUDY

## A.1 SCORING MODEL

The automatic evaluation at the Generation level uses a standardized prompt to guide the scoring models. This prompt, shown in the box below, instructs the evaluator to rate model answers on a scale from 0 to 1 based on specific rules.

---

**Prompt Templates for OpenEvaluator**

"You are an expert evaluator. Given the following question, reference answer, and model answer. Rate the model answer on a scale of 0-1 and provide a BRIEF explanation.",

**Scoring Rules:**

**1. Modality:**
- Reference answers are images. Model outputs may be images or text descriptions.
- If the model outputs an image (matching modality): max score 0.8.
- If the model outputs text (different modality): text descriptions are *valid* and should be evaluated, max score 0.8.
- Many models cannot output images; text descriptions are acceptable alternatives.

**2. Scale (0.1 increments):**
- 0.0: Incorrect, irrelevant, or fails to address the question.
- 0.1–0.2: Mostly incorrect, minimal relevance.
- 0.3–0.4: Partially correct, significant errors or omissions.
- 0.5–0.6: Moderately correct, missing key information.
- 0.7–0.8: Mostly correct, minor errors.
- 0.9–1.0: Excellent to perfect (1.0 reserved for flawless answers with matching modality).

**3. Evaluation Criteria (weights: Correctness 60%, Completeness 25%, Relevance 15%):**
- **CRITICAL:** Focus on *final answer* accuracy, not the reasoning process.
- **CONCISENESS:** Long explanations do not earn higher scores.
- **STRICTNESS:** High scores (0.7+) are only for genuinely accurate and complete answers.

**4. Guidelines:**
- Be **strict**: high scores (0.7) only for genuinely accurate answers.
- Wrong final answer $\rightarrow$ low score regardless of reasoning.
- Use discrete scores: 0.0, 0.1, 0.2, ..., 1.0.

"Output format: \\**score{X}** where X [0.0, 1.0]."

**Question:** f"Question: {question}"

---

To assess the robustness and evaluator-independence of our scoring protocol, we evaluate the same set of model-generated outputs using multiple scoring models. Our evaluation includes GPT-4o (with two independent runs for test–retest validation), Intern-S1, and Claude-Opus-4. Applying these evaluators to the identical generation set allows direct, controlled comparison of scoring consistency across models.

Figure 6 presents the Pearson correlation matrix across all evaluator pairs. All off-diagonal correlation coefficients exceed $r > 0.91$ (mean $r = 0.939$, SD $= 0.014$), indicating extremely high

consistency among evaluators. Notably, the two independent runs of GPT-4o achieve the highest correlation ($r = 0.962$), demonstrating strong test–retest reliability.

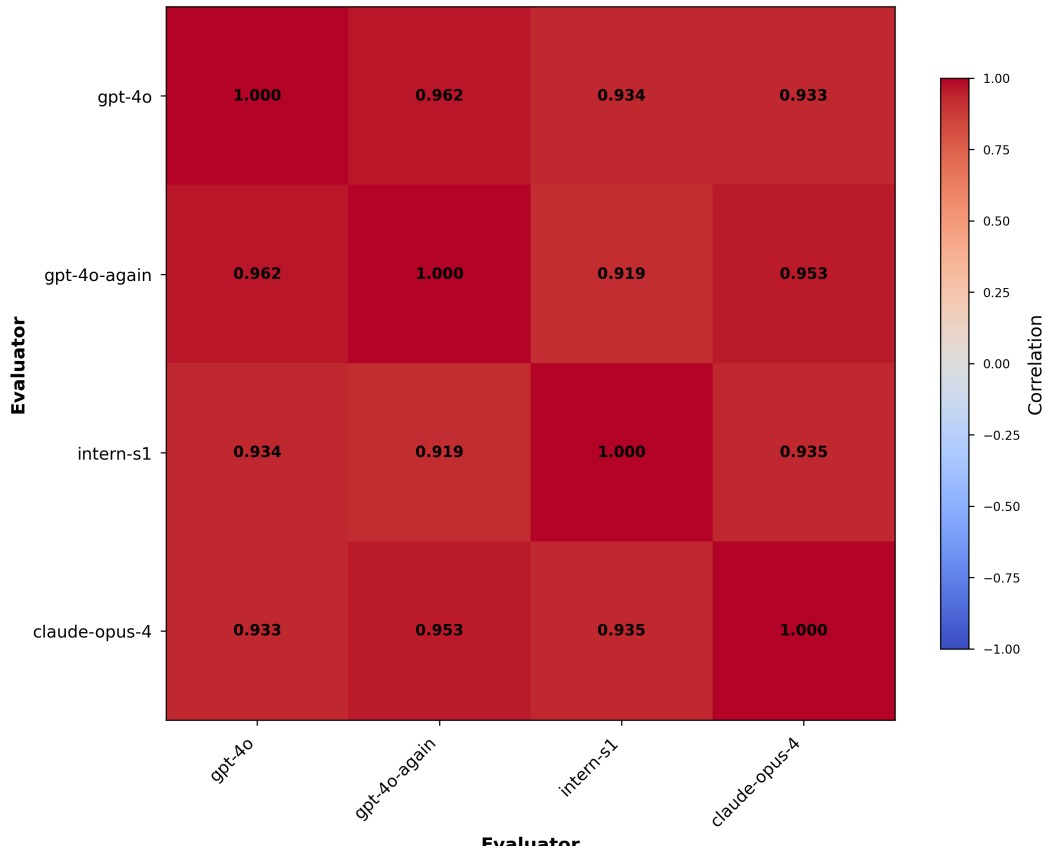

Figure 6: **Evaluator Correlation Matrix.** Pearson correlation coefficients between all scoring models. Higher values (darker colors) indicate stronger agreement.

Figure 7 further visualizes pairwise evaluator agreement using scatter plot matrices. The scatter plots exhibit highly linear alignment along the $y = x$ diagonal for all evaluator pairs, confirming that different scoring models produce nearly identical score distributions. Minor deviations appear consistently across models.

**Together, these results show that our evaluation protocol is stable, evaluator-agnostic, and does not rely on any specific LLM.**

A.2 TEMPERATURE

| Temp | Qwen2.5 VL-32B | | | | Qwen2.5 VL-72B | | | | InternVL3-78B | | | |
|------|--------|------------|----------|------------|--------|------------|----------|------------|--------|------------|----------|------------|
| | Signal | Perception | Semantic | Generation | Signal | Perception | Semantic | Generation | Signal | Perception | Semantic | Generation |
| 1.0 | 57.02 | 48.89 | 17.03 | 6.67 | 78.00 | 72.49 | 78.35 | 13.51 | 69.18 | 73.33 | 71.84 | 9.53 |
| 0.5 | 63.40 | 68.63 | 29.49 | 36.23 | 66.49 | 65.36 | 58.97 | 17.39 | 66.49 | 65.36 | 58.97 | 17.39 |
| 0.0 | 63.92 | 70.59 | 28.85 | 37.68 | 63.92 | 60.78 | 66.03 | 17.39 | 63.92 | 60.78 | 66.03 | 17.39 |

Table 4: Performance of three models under varying temperature settings.(Top-$p$ fixed to 1)

Table 4 presents the impact of varying temperature settings on three models: Qwen2.5-VL-32B, Qwen2.5-VL-72B, and InternVL3-78B.

For Qwen2.5-VL-32B, lower temperatures ($T = 0.5$ and $T = 0$) yield substantial improvements over $T = 1.0$, particularly on the Perception, Semantic, and Generation levels. A similar trend

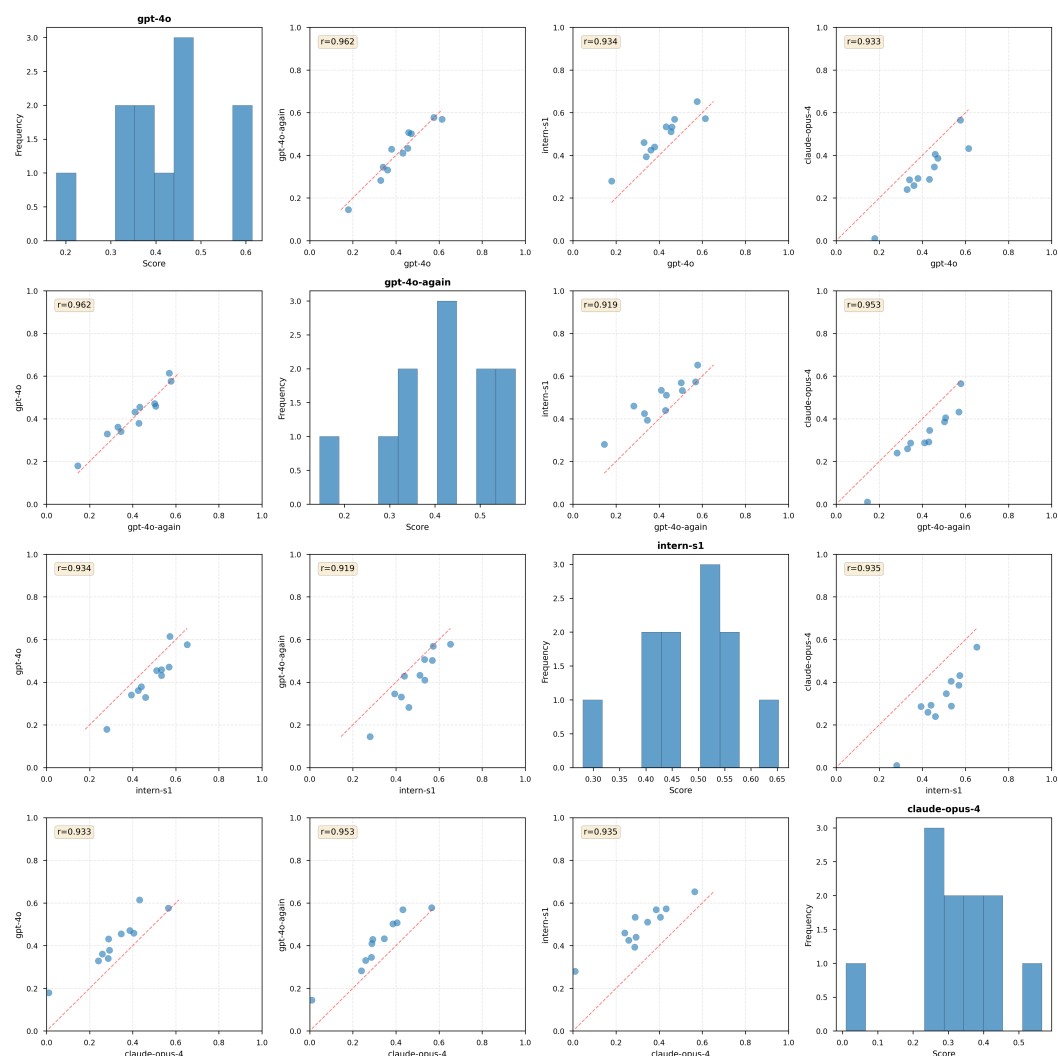

Figure 7: **Pairwise Evaluator Agreement.** Scatter plot matrix comparing the scores produced by different evaluators. Each off-diagonal cell shows a pairwise comparison, with the diagonal showing score distributions.

is observed for InternVL3-78B, where deterministic decoding ($T = 0$ or $T = 0.5$) leads to a more balanced performance profile compared to the stochastic setting. In contrast, Qwen2.5-VL-72B behaves differently: while it achieves the highest Signal and Semantic scores at $T = 1.0$, its Generation accuracy remains relatively low across all settings.

These observations indicate that smaller models tend to benefit from reduced sampling variability, as lower temperatures enhance stability and reliability. Conversely, larger models may require higher temperatures to fully exploit their expressive capacity, though this comes at the cost of weaker generative consistency.

| Top-$p$ | Qwen2.5 VL-32B | | | | Qwen2.5 VL-72B | | | | InternVL3-78B | | | |
|---|---|---|---|---|---|---|---|---|---|---|---|---|
| | Signal | Perception | Semantic | Generation | Signal | Perception | Semantic | Generation | Signal | Perception | Semantic | Generation |
| 1.0 | 57.02 | 48.89 | 17.03 | 6.67 | 78.00 | 72.49 | 78.35 | 13.51 | 69.18 | 73.33 | 71.84 | 9.53 |
| 0.5 | 63.92 | 69.93 | 29.49 | 39.13 | 62.89 | 62.09 | 65.38 | 13.04 | 70.62 | 68.63 | 72.44 | 5.80 |
| 0.1 | 63.40 | 66.67 | 30.77 | 44.93 | 66.49 | 54.25 | 64.74 | 23.19 | 69.07 | 69.93 | 72.44 | 11.59 |

Table 5: Performance of three models under varying Top-$p$ settings.(temperature fixed to 1).

## A.3 TOP-$p$

We further investigate the role of nucleus sampling while fixing the temperature to 1.0. The results in Table 5 show heterogeneous effects across models.

For Qwen2.5-VL-32B, reducing Top-$p$ from 1.0 to 0.1 consistently improves Semantic and Generation scores, suggesting that constraining the sampling space mitigates low-quality outputs and enhances reliability. By contrast, Qwen2.5-VL-72B attains its best Signal and Semantic results at $p = 1.0$, but its Generation score is substantially reduced. Interestingly, setting $p = 0.1$ recovers part of this loss, implying a trade-off between precision and diversity.

For InternVL3-78B, performance remains comparatively stable across Top-$p$ values, with minor fluctuations in Generation accuracy. This stability suggests that larger-scale models are less sensitive to sampling truncation, reflecting stronger intrinsic consistency.

## B TASK HIERARCHY

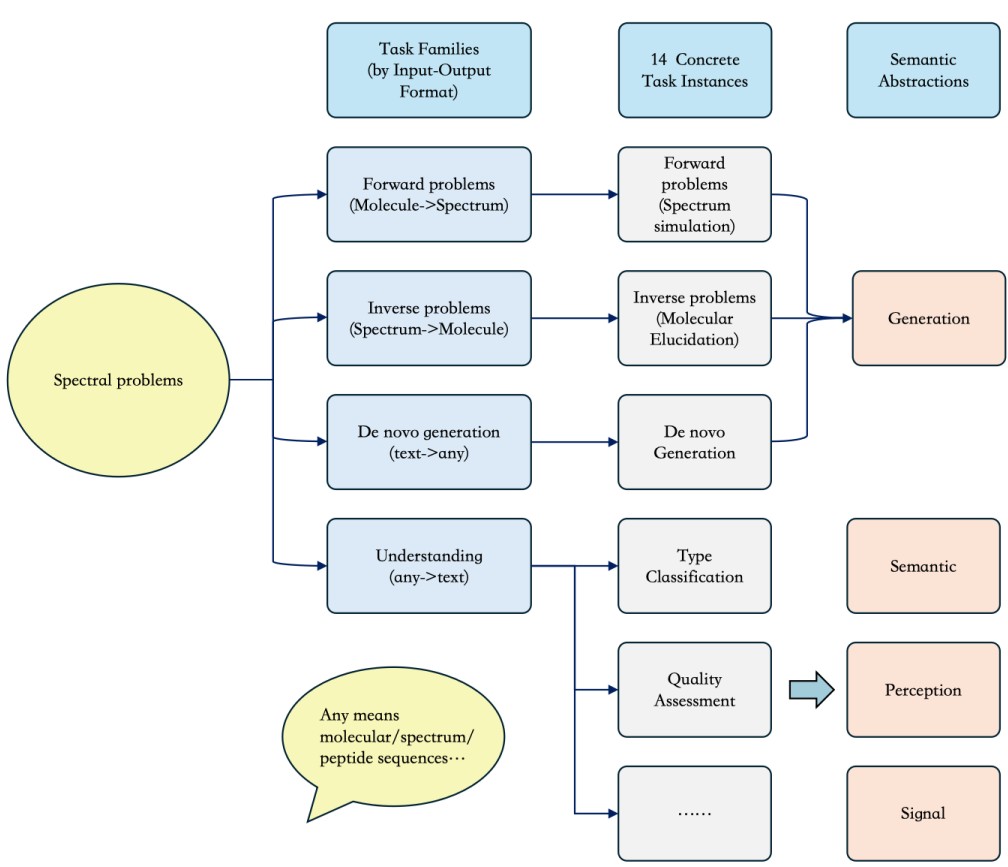

Figure 8: **Task Hierarchy of Our Benchmark.** The hierarchical structure of tasks spanning different difficulty levels, modalities, and evaluation criteria.

Figure 8 summarizes the construction of the SpectrumBench taxonomy. Starting from a broad view of spectrum-related problems (left side), we organize all tasks according to their input–output structure, forming four high-level families: (1) forward problems (molecule → spectrum), (2) inverse problems (spectrum → molecule), (3) de novo generation (text → molecular, spectral, or peptide sequences), and (4) understanding tasks (any → text).

Each of these families is then instantiated as the 14 concrete sub-tasks included in SpectrumBench (shown in the middle of the figure). These sub-tasks are further abstracted into four semantic lev-

els—*Signal*, *Perception*, *Semantic*, and *Generation*—which are consistently used throughout the paper.

In spectroscopy, predicting a spectrum from a known molecular structure is typically categorized as a forward problem: the molecular configuration (geometry, bonding, electronic structure) determines the physico-chemical state, while the observed spectrum is a compressed manifestation of that state. This view is consistent with standard formulations of forward modelling in computational chemistry and spectroscopy. Conversely, recovering a molecular structure from spectral measurements constitutes an inverse problem: the task seeks to reconstruct a richer latent representation from partial, noisy, and sometimes ambiguous observations. Lu et al. (2025)

We make this distinction explicit to ensure clarity for ML readers, as the forward–inverse terminology is widely used in the spectroscopy and inverse-problem literature but may be unfamiliar outside these communities.

## C SPECTRUMBENCH DETAILED INFORMATION

### C.1 EXPLORATORY DATA ANALYSIS

To ensure the transparency, reliability, and scientific relevance of SpectrumBench, we conducted a comprehensive exploratory data analysis (EDA) of all datasets included in SpectrumBench. This analysis summarizes the provenance, spectral modalities, molecular coverage, and data-generating processes (experimental or computed) for each dataset. Our goal is to provide a clear characterization of the benchmark's multimodal landscape and to help practitioners understand the heterogeneity of real-world spectroscopic data.

Table 6 presents the EDA summary across ten datasets. For each dataset, we report: (1) Source Type (public or in-house), (2) Modalities Included (e.g., NMR, MS, IR, Raman, SMILES), (3) Number of Molecules, (4) Spectrum Origin (experimental or computed), and (5) A brief description capturing data provenance and major properties.

These datasets span diverse use cases, from quantum-chemistry–based simulated spectra (QM9S, Private Lab) to large-scale experimental NMR collections (NMRBank, NMRShiftDB, SupportInfo-Crawl), peptide MS/MS corpora (NovoBench), and curated textbook-level structure elucidation tasks (MolPuzzle). Importantly, many datasets include multiple modalities per molecule, reflecting natural co-occurrence patterns observed in real analytical workflows. This multimodal overlap further supports the design choice of unifying spectroscopy tasks within a single benchmark framework.

### C.2 JUSTIFICATION FOR A MULTIMODAL BENCHMARK

Spectroscopic analysis in practice is intrinsically multimodal: a single molecule is frequently characterized by MS/MS, NMR, IR, Raman, and a symbolic structure representation (e.g., SMILES). SpectrumBench therefore aims to evaluate not only single–modality reasoning, but also a model's ability to integrate these complementary signals.

Table 7 reports the multimodal coverage across the core datasets used in SpectrumBench. QM9S, the Multimodal-Spectroscopic-Datasets, and MolPuzzle provide fully paired modalities for every molecule, supporting clean multimodal training. In contrast, SDBS exhibits partial pairing across modalities; Table 8 quantifies this variation (e.g., IR covers 91% of molecules, ESR only 2%).

These observations motivate a multimodal benchmark for two reasons: (i) many real datasets naturally contain paired modalities, making multimodal learning realistic rather than synthetic; and (ii) incomplete pairing, as seen in SDBS, reflects common real-world scenarios where cross-modal reasoning is crucial. For these reasons, multimodality is treated as a first-class design principle in SpectrumWorld.

### C.3 SIGNAL LEVEL

This layer focuses on the direct processing and understanding of raw, fundamental data formats, much like extracting information from physical signals, as exemplified in Figure 9.

| Dataset | Source | Modalities | #Molecules | Origin | Description |
|---|---|---|---|---|---|
| QM9S2023a | Public | SMILES, UV, IR, Raman | 130,000 | Computed | Derived from the QM9 small-molecule set, QM9S contains 130,831 synthetically feasible organic molecules with up to nine heavy atoms (C, N, O, F). UV, IR and Raman spectra are computed via high-level quantum-chemical simulations for each molecule. |
| Multimodal-Spectroscopic-Datasets 2024b | Public | SMILES, $^1$H NMR, $^{13}$C NMR, HSQC-NMR, MS$^+$, MS$^-$, IR | 790,000 | Computed | This dataset includes simulated multi-modal spectra for approximately 790,000 molecules extracted from a large patent-reaction corpus, providing paired NMR, MS (positive/negative) and IR spectra for each structure. |
| MolPuzzle2024b | Public | SMILES, IR, MS, $^1$H NMR, $^{13}$C NMR | 217 | Experimental | Derived from 217 unique molecule-elucidation problems curated from a chemistry textbook, where each molecule is paired with IR, MS, $^1$H and $^{13}$C NMR spectra to support structure reasoning. |
| NMRBank 2025c | Public | SMILES, $^1$H NMR, $^{13}$C NMR | 225,809 | Experimental | A large-scale experimental NMR database built via automated extraction from roughly 5.73 million open-access scientific publications. Each entry includes IUPAC name, SMILES (for a large subset), experimental conditions (solvent, field strength), $^1$H/$^{13}$C chemical shifts, and confidence scores. |
| NovoBench2024a | Public | SMILES, MS/MS spectra | N/A | Experimental | NovoBench aggregates three widely used MS/MS resources for de novo peptide sequencing: a Seven-Species collection of low-resolution spectra from seven organisms, a Nine-Species collection of high-resolution spectra from nine species including common PTMs, and the HC-PT set of high-confidence human peptide spectra. Together they provide paired MS/MS spectra and peptide labels for standardized benchmarking. |
| MassSpecGym2024a | Public | SMILES, MS/MS spectra | ~29,000 | Experimental | MassSpecGym aggregates 231,104 high-quality tandem MS (MS/MS) spectra corresponding to 28,995 unique molecular structures. Spectra are sourced from public spectral libraries such as GNPS, MoNA and MassBank, as well as additional in-house measurements, and each spectrum is linked to a canonical SMILES string with standardized acquisition metadata. |
| NMRShiftDB2024 | Public | SMILES, $^1$H NMR, $^{13}$C NMR | ~40,000 | Experimental | NMRShiftDB contains experimental $^1$H and $^{13}$C NMR spectra collected from the literature and community submissions. All entries undergo structural validation and manual curation, making it one of the largest open databases of real NMR chemical shifts. |
| SDBS | Public | SMILES, IR, MS, $^1$H NMR, $^{13}$C NMR, Raman, ESR | 43,628 | Experimental | The SDBS dataset aggregates 43,628 experimental spectral records from the Spectral Database for Organic Compounds maintained by AIST (Japan). For each compound, it provides IR, MS, $^1$H/$^{13}$C NMR, Raman or ESR spectra, together with detailed measurement conditions (solvent, field strength, sample preparation, laser parameters, reaction conditions) and standardized metadata such as molecular formula, molecular weight, compound name and CAS registry number. |
| Private Lab Dataset | In-house | SMILES, IR, Raman | 417,012 | Computed | An in-house quantum-chemistry dataset containing IR and Raman simulations for 417,012 molecules with 3D atomic coordinates and electronic properties. Each record includes a canonical SMILES string, 3D coordinates, Hartree–Fock energy, dipole moment and derivatives, polarizability and polarizability derivatives for molecules ranging from 2 to 70 atoms. |
| SupportInfo-Crawl | To be released | SMILES, $^1$H NMR, $^{13}$C NMR, MS, $^{19}$F/$^{11}$B/$^{31}$P NMR | ~320,000 | Experimental | Our crawled NMR dataset contains experimentally measured spectra (primarily $^1$H and $^{13}$C NMR) extracted from chemical literature via a custom end-to-end pipeline that retrieves Supporting Information using DOI links and performs automated spectrum extraction and scaling. From 388,831 papers, we obtained about 320,000 valid molecular entries (around six per article); most molecules have paired $^1$H/$^{13}$C NMR spectra, and additional modalities such as MS, $^{19}$F, $^{11}$B and $^{31}$P NMR are also covered. |

Table 6: **Exploratory Data Analysis (EDA) summary of all datasets included in Spectrum-Bench.** For each dataset we report its source type, included modalities, molecular scale, spectrum origin, and a brief description.

## C.4 PERCEPTION LEVEL

This layer associates the features identified at the signal layer with chemical entities (functional groups, fragments, elements, and basic properties), as illustrated in Figure 10.

## C.5 SEMANTIC LEVEL

This layer involves higher-level reasoning and comprehensive interpretation, connecting fragmented information to form complete insights or generate novel chemical structures, as depicted in Figure 11.

| Dataset | Modalities Included | Fully Paired? | % Molecules with Complete Set |
|---|---|---|---|
| QM9S | UV, IR, Raman | Yes | 100% |
| Multimodal-Spectroscopic-Datasets | C-NMR, H-NMR, HSQC-NMR, MS$^+$, MS$^-$, IR | Yes | 100% |
| MolPuzzle | IR, MS, HNMR, CNMR | Yes | 100% |
| SDBS | IR, MS, HNMR, CNMR, Raman, ESR | No | Variable (2–91%) |

Table 7: Multimodal completeness and modality pairing across datasets in SpectrumWorld. "Fully paired" indicates that each molecule contains all listed modalities.

| Modality | #Spectra | % Molecules Containing This Modality |
|---|---|---|
| IR | 39,980 | 91.64% |
| MS | 32,838 | 75.27% |
| HNMR | 18,317 | 41.98% |
| CNMR | 17,688 | 40.54% |
| Raman | 4,569 | 10.47% |
| ESR | 924 | 2.12% |

Table 8: Distribution of spectral modalities within the SDBS dataset. Each row reports the number of spectra and the percentage of molecules containing that modality.

### C.6 GENERATION LEVEL

This layer focuses on creating novel data, such as generating a 2D image of a molecule from its SMILES string, predicting the Mass Spectrum for a given chemical structure, or designing a new molecule with specific properties, as illustrated in Figure 13.

### C.7 DATA DISTRIBUTION

To provide an overview of the data landscape, Figure 12 presents two pie charts: the left illustrates the distribution of different spectrum types (*e.g.*, NMR, IR), while the right shows the categorization of spectroscopic task types. These distributions reflect the diversity of data and tasks within our study. It should be noted that the spectrum type statistics were generated by having GPT-4o scan and summarize all spectra in the benchmark. However, there are potential limitations: GPT may have recognition errors, and some spectrum-involving benchmarks lack actual image data (e.g., predicting NMR spectrum properties from molecular characteristics in *de novo* generation tasks). Additionally, in tasks like multimodal fusion reasoning and forward generation problems, a single benchmark instance might include multiple spectra. Thus, the number of spectra does not align with the number of benchmarks, and this pie chart is provided only as a general reference.

## D SPECTRUMANNOTATOR TECHNICAL DETAILS

In the main text, we briefly introduced the function of SpectrumAnnotator. In this section, we will introduce its specific technical details.

MolPuzzle (Guo et al., 2024b) represents the first benchmark specifically designed for LLMs in spectroscopic analysis, employing a three-stage approach to generate question-answer pairs. While this template-based generation method offers efficiency, it suffers from limited coverage of spectroscopic domains and overly simplistic question formats. In the field of spectroscopy, high-quality data and benchmarks are crucial to advance AI research. The design of SpectrumAnnotator originates from two key insights: First, the process of creating benchmarks shares similarities with the supervised data generation methods used in LLM pre-training and post-processing. Just as high-quality training data is essential for model performance, well-designed benchmarks are equally critical for evaluating and advancing the field. Second, we aim to utilize LLMs' few-shot and zero-shot capabilities to generate diverse benchmarks, enabling batch processing of seed datasets to construct large-scale pre-training and post-processing data. Additionally, we leverage LLMs' discriminative abilities for preliminary data screening and establish closed-loop mechanisms for continuous improvement.

| | Examples |
| --- | --- |
| | Identifying the type of a spectrum, assessing its data quality, extracting basic features (e.g.,peak position, peak intensity), and identifying impurity peaks.. |

| Sub-Category | Metadata |
| --- | --- |
| Spectrum Type Classification | **Question:**
What type of spectrum is this?
**Choices & Answer:**
A. Infrared Spectrum (IR).
B. Proton Nuclear Magnetic Resonance (H-NMR).
C. Heteronuclear Single Quantum Coherence (HSQC).
D. Raman Spectrum.
**Explanation:**
The spectrum uses ppm as units, which is a chemical shift unit specific to NMR. The chemical shift range typically falls between -2 ppm and 15 ppm, confirming this is a 1H NMR spectrum. |
| Spectrum Quality Assessment | **Question:**
Does this spectrum show obvious signal quality issues?
**Choices & Answer:**
A. Yes.
B. No, the signal is very clear.
C. Localized noise.
D. Very low noise, egligible.
**Explanation:…** |
| Basic Feature Extraction | **Question:**
Please select the chemical shift range corresponding to the most concentrated signal area in the HSQC spectrum.
**Choices & Answer:**
A. $\delta$H 2-4 ppm, $\delta$C 30-60 ppm.
B. $\delta$H 6-8 ppm, $\delta$C 120-140 ppm.
C. $\delta$H 9-10 ppm, $\delta$C 180-200 ppm.
D. $\delta$H 0-1 ppm, $\delta$C 10-20 ppm.
**Explanation:**
HSQC spectrum plots $^1$H chemical shift on the horizontal axis and $^{13}$C on the vertical. Most signals cluster in the 2-4 ppm ($^1$H) and 30-60 ppm ($^{13}$C) region. |
| Impurity Peak Detection | **Question:**
Please observe this spectrum carefully. Besides the signals from the target compound, there is also a distinct additional peak around 1 ppm in the image. What is this peak most likely?
**Choices & Answer:**
A. Solvent impurity.
B. Target compound.
C. Instrument noise.
D. Reference standard.
**Explanation:**
In NMR spectrum, the peak near 1 ppm is often from impurities introduced during sample processing. Given it's an "extra" signal not part of the target compound, it's likely an impurity. |

Figure 9: Example tasks and question formats at the Signal Level.

As illustrated in Figure 14, SpectrumAnnotator consists of several key components that work together to generate high-quality spectroscopic benchmarks. **Configuration & Seed Datasets** form the foundation of the system. Seed datasets are extracted from multiple data sources containing essential spectroscopic information, while the configuration is a YAML configuration file that primarily configures prompt templates, instructing the generator on what prompts to use, along with model configurations and other parameters. As shown in Figure 15, taking property prediction as an example, the configuration specifies the seed datasets from MolPuzzle and provides question templates to guide the generator's output.

**DataLoader** addresses the challenge of integrating diverse data sources. Ideally, we would like to standardize all seed datasets into a uniform format. However, in practice, this proves challenging as original data may possess complex nested file structures and diverse storage formats. To reduce adaptation complexity, we allow customized DataLoader designs. This design is inspired by Py-

| Examples |
| --- |
| Identifying functional groups like -OH from a mass spectrum; determining the presence of isotopes like $^{13}C$; assigning a $^1H$ NMR triplet to a methyl group; predicting molecular weight from a mass spectrum. |

| Sub-Category | Metadata |
| --- | --- |
| Basic Property Prediction | **Question:** Given the mass spectrum image, what is the most likely molecular ion peak (m/z) observed for this compound? **Choices & Answer:** A. 85. B. 107. C. 120. D. 150. **Explanation:** The strongest peak at m/z 107.0 is the molecular ion (M+), with an adjacent m/z 109.0 peak (~1/3 intensity) indicating one chlorine atom ($^{35}Cl/^{37}Cl \approx 3:1$). Smaller peaks (m/z 93.0, 108.0) are fragments. |
| Elemental Composition Prediction | **Question:** Observe the provided mass spectrum image. The significant M+2 peak suggests the presence of which element? **Choices & Answer:** A. Fluorine (F). B. Chlorine (Cl). C. Bromine (Br). D. Iodine (I). **Explanation:** The intensity ratio of the m/z 51 and 53 peaks (~3:1) reflects chlorine's natural isotopes, $^{35}Cl$ (75.77%) and $^{37}Cl$ (24.23%), giving an M+2 peak about one-third the main peak. |
| Functional Group Recognition | **Question:** Based on this infrared spectrum, what functional group is most likely present in the molecule? **Choices & Answer:** A. Carbonyl group (C=O). B. Hydroxyl group (-OH). C. Amino group (-NH2). D. Nitro group (-NO2). **Explanation:** In the infrared spectrum, a pair of sharp absorption peaks around 3300 cm$^{-1}$ are typical of the symmetric and asymmetric N–H stretching vibrations in a primary amino group ($-NH_2$). |
| Peak Assignment | **Question:** Given the chemical formula C6H5F. Observe this H-NMR spectrum. The singlet peak around ~7.3 ppm in the image is most likely assigned to which part of the molecule? **Choices & Answer:** A. Methyl group. B. Fluoro-substituted carbon. C. Aromatic ring protons. D. Alkene protons **Explanation:** The 7.3 ppm shift is typical for aromatic protons in fluorobenzene ($C_6H_5F$). Though misdescribed as a singlet, it's a complex multiplet from H-H and H-F coupling, with the shift confirming its aromatic nature. |

Figure 10: Example tasks and question formats at the perception level.

Torch's DataLoader, which can properly load, batch, and post-process raw data. Our DataLoader aims to integrate various "seed datasets" into formats that can be processed by generators. The foundation consists of two base classes: DataSample, which represents the minimal granular information unit in SpectrumAnnotator and serves as reference information for the Generator to generate individual samples; and Dataset, a collection of DataSample objects that provides standardized access methods. As demonstrated in Figure 16, the DataLoader adopts a plugin-based architecture with an abstract registry. For different seed datasets, researchers only need to register their custom loaders using simple registration code, enabling seamless integration of diverse data sources.

| Examples |
| Elucidating a complete molecular structure from one or more spectra; verifying a proposed structure against spectral data; and reasoning across different modalities (e.g., text and spectrum) to answer complex questions. |

| Sub-Category | Metadata |
|---|---|
| **Fusing Spectroscopic Modalities** | **Question:**
The molecular formula of the compound is C6H11NO. Use this information together with the provided IR spectrum to infer possible structural features.
**Choices & Answer:**
A. Amide.
B. Alcohol.
C. Ester.
D. Alkene.
**Explanation:**
Infrared spectroscopy shows a strong 1650 cm⁻¹ peak (C=O) and a 3300–3500 cm⁻¹ peak (N–H). Their coexistence, along with N and O in the formula, clearly indicates an amide group. |
| **Molecular Structure Elucidation** | **Question:**
Given the mass spectrum of an unknown compound with a molecular formula C11H16, predict the most likely molecular structure (SMILES) consistent with the observed fragments.
**Choices & Answer:**
A. CC(C)=C1C=CC=CC1.
B. CC(C)CC1=CC=CC2=CC=CC=C12.
C. CC(C)(C)CC1=CC=CC=C1.
D. CCC(C)C1=CC=CC2=CC=CC=C12.
**Explanation:**
The base peak at m/z 91 indicates a benzyl (C₆ H₅ CH₂ –) structure, while m/z 133 represents loss of a methyl group. Only CC(C)(C)CC1=CC=CC=C1 fits both fragmentations. |
| **Multimodal Molecular Reasoning** | **Question:**
The Raman spectrum of the molecule OC1CCC1=O (2-hydroxycyclopentanone) shows a series of strong peaks in the 2800-3000 cm⁻¹ region. These peaks are most likely attributed to which type of molecular vibration?
**Choices & Answer:**
A. C-H stretching.
B. O-H stretching.
C. C=O stretching.
D. N-H stretching.
**Explanation:**
In Raman spectroscopy, 2800–3000 cm⁻¹ is characteristic of C–H stretching. The strong peak here arises from cycloalkane C–H vibrations, while O–H (3200–3600 cm⁻¹) and C=O (~1700 cm⁻¹) peaks are absent. |

Figure 11: Example tasks and question formats at the semantic level.

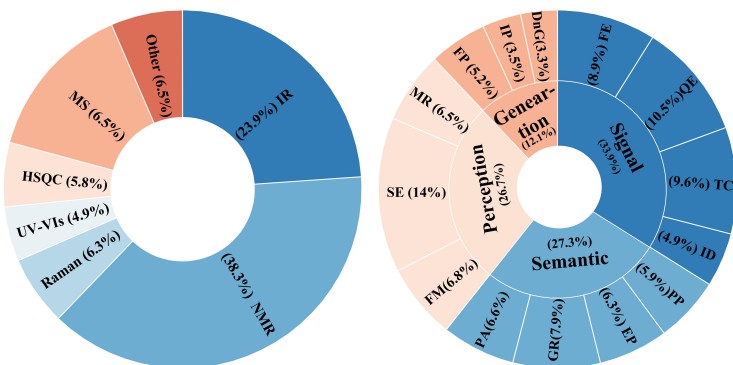

Figure 12: Distribution of spectrum types and spectroscopic task categories.

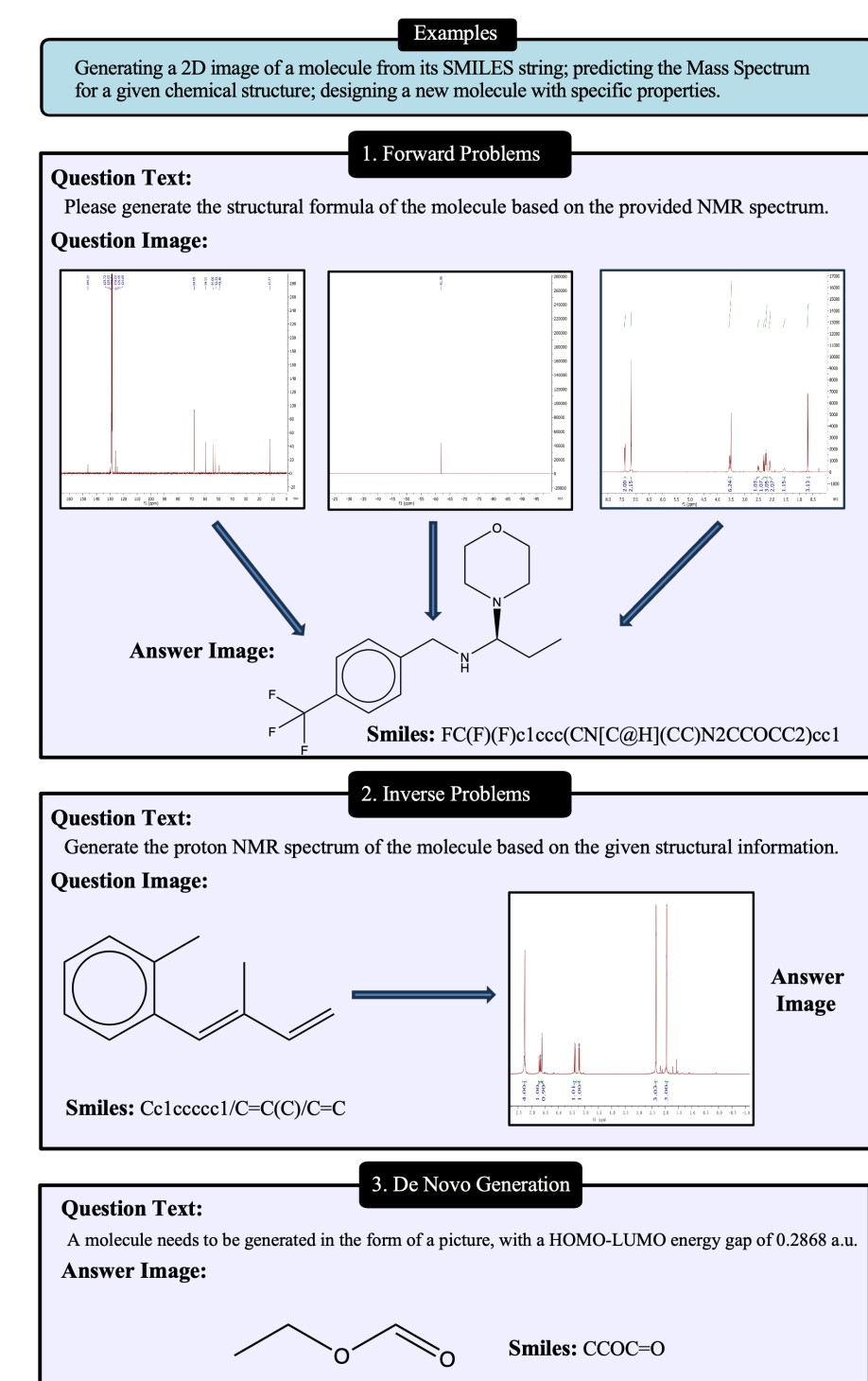

Figure 13: Example tasks and question formats at the Generation Level.

**Generator** operates through a three-stage workflow: First, it receives question templates from Configuration (including few-shot examples). Second, for each sample in the seed dataset, the generator uses question templates combined with sample metadata (such as molecular formulas, spectrum paths, SMILES strings, etc.) to render a prompt, which is then passed to the large language model. Third, the model's output is parsed into standard formats (e.g., question/choices/answer).

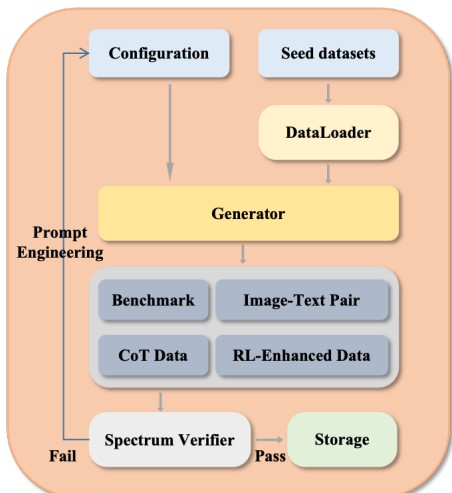

Figure 14: Technical architecture of SpectrumAnnotator, illustrating the data flow from seed datasets through generation to quality verification.

---

**Prompt Template**

dataset_path: seed_datasets/molpuzzle/meta_data.json

generator_type: benchmark

model_type: internvl

num_samples: 30

output_path: output/benchmark.json

property_pred:

    template: |

        This is a property prediction task. You are given the following spectrum(s) for a molecule: {spectra_list}

        You are provided with one or more spectrum images (such as MS, IR, NMR, UV-Vis, etc.) for a molecule.

        Your task is to create a question and answer that require the examinee to infer a chemical or physical property of the compound based on the features visible in the provided spectrum image(s).

        Typical properties include: molecular weight (from mass spectrum), acidity (from IR), color (from UV-Vis), solubility (from NMR), and so on. Please focus on designing questions that require the examinee to deduce such properties from the actual spectral features shown in the image(s).

        Here are some example questions for your reference. Please output your result in the following JSON format (making sure image_path points to the actual spectrum image and is not empty):

        Example 1:

        {{

         "selected_spectrum_type": "MASS",

         "question": "Given the mass spectrum image, what is the most likely molecular ion peak (m/z) observed for this compound?",

         "choices": ["85", "90", "120", "150"],

         "answer": "120",

        }}

        Example 2:

        {{

         "selected_spectrum_type": "IR",

         "question": "Given the infrared spectrum image, a very broad absorption band in the 2500−3300 cm⁻¹ range and a strong peak around 1700 cm⁻¹ are observed. What property does this most likely suggest for the compound?",

         "choices": ["Strong acidity", "High basicity", "Aromaticity", "Aliphatic character"],

         "answer": "Strong acidity",

         }}

Figure 15: Example configuration for property prediction tasks, demonstrating how prompt templates and model parameters are specified.

```
                        DataLoader

    @dataclass
    class DataSample:
       id: str
       text: Optional[str] = None
       images: Optional[List[str]] = None
       metadata: Dict[str, Any] = field(default_factory=dict)

    @dataclass
    class Dataset:
       samples: List[DataSample]
       metadata: Dict[str, Any] = field(default_factory=dict)
       def __len__(self) -> int:
          pass
       def __getitem__(self, index: int) -> DataSample:
          pass

    from .base import BaseDataLoader
    @register_loader
    class MoleculeDataLoader(BaseDataLoader):
       def can_handle(self, data_path: Path) -> bool:
          pass
       def load(self, data_path: Path) -> Dataset:
          pass
```

Figure 16: Plugin-based DataLoader architecture showing the registration mechanism for custom data loaders.

**Quality Assurance Pipeline** ensures the reliability of generated benchmarks. After data generation, the system employs a multi-stage quality assurance process: Initial screening using rule-based methods to check data format and remove non-compliant samples, followed by SpectrumVerifier, a large model-based verification system that identifies suspicious samples requiring manual annotation. This closed-loop mechanism ensures that only high-quality, scientifically valid benchmarks are included in the final dataset. SpectrumAnnotator will be open-sourced to collaborate with the research community in building a robust ecosystem and collectively addressing challenges in spectroscopic data generation and curation.

# E  QUANTITATIVE EVALUATION OF ANNOTATION QUALITY

SpectrumBench adopts a mixed automatic–human curation workflow. In early iterations, the SpectrumAnnotator prompts and filtering rules produced a non-trivial amount of noisy or underspecified samples. We therefore carried out several rounds of prompt redesign, rule refinement, and human-in-the-loop verification, and here report the corresponding quantitative evidence.

Figure 17 compares automatic annotations from SpectrumAnnotator with independent manual annotations on a stratified sample of tasks across the four semantic levels (Signal, Perception, Semantic, Generation). For each task, we plot the number of samples judged correct by human annotators and by the automatic pipeline. Overall, the distributions are well aligned, indicating that the automatic pipeline can approximate human labeling quality at scale.

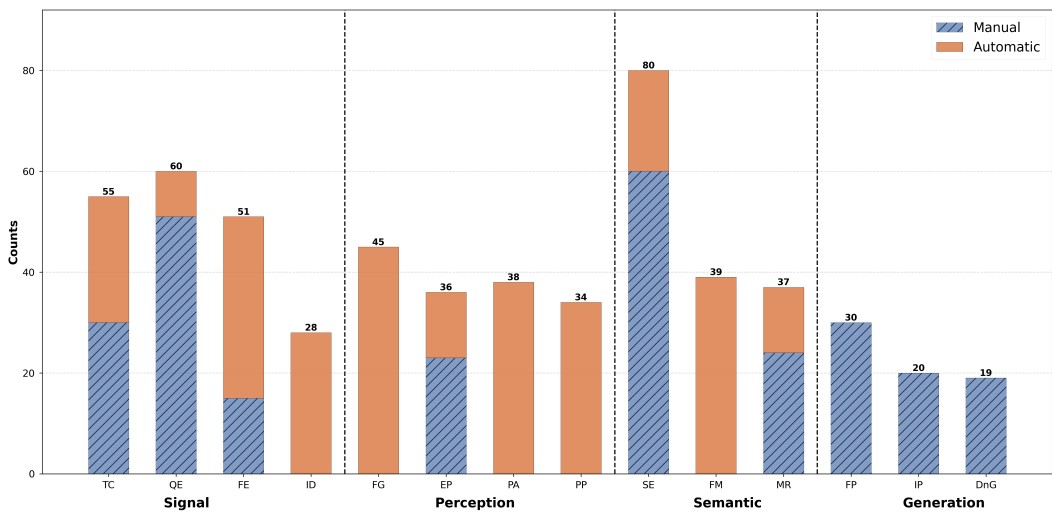

Figure 17: Comparison between automatic (SpectrumAnnotator) and manual annotations across all SpectrumBench tasks. Bars report, for each task type and semantic level, the number of samples judged correct by human annotators (blue) and by the automatic pipeline (orange).

To further quantify annotation quality, Table 9 reports the aggregate error rates of the *automatically generated* annotations (before human verification) across the three semantic levels. Signal and Perception tasks show reductions from $57.4\%$ to $21.2\%$ and from $52.8\%$ to $19.8\%$, respectively, while Semantic tasks improve from $74.6\%$ to $26.2\%$. Generation tasks are excluded because they are fully manual annotation and therefore do not rely on automatic annotation.

These results demonstrate that iterative refinement, substantially improves the reliability of *automatic* annotations. Importantly, the reported error rates refer only to raw auto-generated outputs; all benchmark data released in SpectrumBench undergo a subsequent human-verification stage, and the final benchmark does not contain these errors.

A further benefit of this workflow is efficiency: instead of manually designing and writing thousands of benchmark items, curators only need to validate and correct model-proposed annotations. This dramatically reduces the human labor required for benchmark construction while preserving scientific rigor.

We also note that the remaining errors after refinement primarily stem from model limitations (our pipeline currently uses Intern-S1). Stronger models would further reduce these residual issues, and for several simpler tasks—such as spectrum-type classification—the refined prompts already achieve near-zero automatic error. This indicates that the annotation pipeline scales with model capability and can continue to improve as foundation models advance.

| Error rate | Signal | Perception | Semantic | Generation |
|---|---|---|---|---|
| Before refinement | 57.4% | 52.8% | 74.6% | N/A |
| After refinement | 21.2% | 19.8% | 26.2% | N/A |

Table 9: Annotation error rates for SpectrumAnnotator before and after iterative prompt and rule refinements, aggregated at each semantic level. Generation tasks are open-ended and are not included in this calculation.

## F  BENCHMARKING CANDIDATES

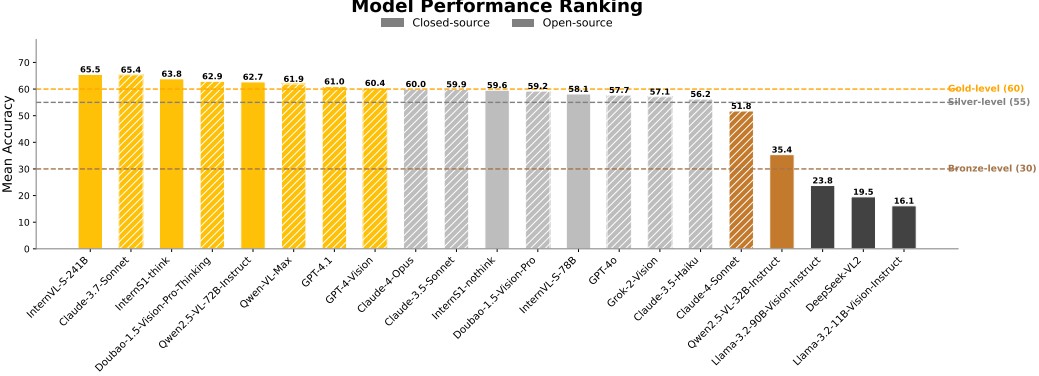

Figure 18: Performance ranking of various LLMs.

### F.1  OPEN-SOURCE MODELS

**Qwen2.5-VL-32B-Instruct(Bai et al., 2025).** Alibaba's open-source Vision-Language multimodal large model that handles reasoning and generation for images, text and video. It employs a hierarchical tagging architecture, supports multi-turn conversations and complex reasoning, and both the model weights and code are publicly available.

**Qwen2.5-VL-72B-Instruct(Bai et al., 2025).** Qwen2.5's larger-scale model enhances cross-modal reasoning and instruction-following capabilities, delivering superior performance on benchmarks such as MMMU and M3Exam while supporting multitasking and multilingual inputs - and is completely open-source.

**InternVL3-78B (Chen et al., 2024b).** Shanghai AI Lab releases the multimodal model, combining native multimodal pre-training, variable visual position encoding (V2PE), MPO, and test-time scaling to approach GPT-4o performance.

**Llama-3.2-11b-Vision-Instruct(Meta AI, 2024).** Meta's 11 B lightweight multimodal model locks Llama-3.1 8 B text and pairs it with a ViT encoder. Two-stage training: image-text alignment then SFT+DPO, using RoPE-2D. Open-source.

**Lllama-3.2-90b-Vision-Instruct(Meta AI, 2024).** The 90B features a more advanced vision adapter with cross-attention layers to inject image features into the LLM core. It is tuned with SFT and RLHF for enhanced performance on complex visual reasoning tasks.

**DeepSeek-VL-2(Wu et al., 2024).** An open-source model from DeepSeek-AI featuring a Mixture-of-Experts (MoE) backbone and a dynamic tiling vision encoder for high-resolution images. It

achieves or exceeds the state-of-the-art performance at the time on benchmarks like MMMU and DocVQA, with its code and weights fully available on GitHub.

**Doubao-1.5-Vision-Pro (Doubao Team, 2025).** It features a dynamic resolution visual encoder and MoE architecture, supporting visual QA, text-image matching, and image description. With billions of parameters, it shows strong generalization across scenarios and is available for self-hosting and fine-tuning.

**Doubao-1.5-Vision-Pro-Thinking (Doubao Team, 2025).** It integrates a "Deep Thinking Mode" and is trained with multi-round Reward Learning and reasoning style training. It excels in scientific, mathematical, and chain-of-thought reasoning. Supports open-source calling and API integration.

**GLM-4.5V(VTeam, 2025).** An open-source vision-language model from Zhipu AI and Tsinghua University that introduces a versatile "thinking paradigm" for enhanced reasoning. It leverages scalable reinforcement learning and supports full-spectrum vision reasoning, including GUI agent operations and code generation from screenshots.

**InternS1(Intern-S1 Team, 2025).**A vision-language model developed by Shanghai AI Laboratory that features a specialized "Thinking" mode for enhanced multi-step reasoning. This mode allows the model to perform a series of self-guided logical steps to solve complex problems, particularly in scientific, mathematical, and logical domains.

## F.2 CLOSED-SOURCE MODELS

**GPT-4o (OpenAI, 2024).** OpenAI's flagship "omni" model natively supports text, audio, and image modalities. Delivers GPT-4-level intelligence with significantly faster response times and enhanced multimodal capabilities.

**GPT-4.1(OpenAI, 2025).** A reinforced version of GPT-4 deployed through the OpenAI API, offering improved handling of complex instructions and logical reasoning; accepts multimodal inputs but is primarily geared toward text-centric tasks.

**GPT-4-Vision(OpenAI, 2023).** A version of GPT-4 equipped with image input capabilities, optimized for understanding images and text and for the generation of conversational content, widely used for image-based Q&A.

**Claude-3.5-Haiku.** Anthropic's fastest and most cost-effective model in the Claude3.5 family—offers very low latency, strong coding and reasoning ability, and often exceeds Claude Opus on intelligence benchmarks despite being lightweight.

**Claude-3.5-Sonnet (Anthropic, 2024).** Anthropic's multimodal large language model has mixed inference capabilities and powerful visual understanding functions. It supports a context of 200K tokens and is skilled in natural writing and code generation.

**Claude-3.7-Sonnet (Anthropic, 2025a).** An evolution of Claude3.5 Sonnet that introduces hybrid reasoning—users can choose between fast modes or step-by-step logical chains; offers strong task flexibility, extended context windows, and deep instruction-following in multimodal settings.

**Claude-4-Opus (Anthropic, 2025b).** Anthropic's flagship model, designed for complex tasks. It boasts a powerful memory architecture and parallel tool invocation capabilities, and integrates with Claude Code, performing exceptionally in coding and reasoning benchmark tests.

**Claude-4-Sonnet (Anthropic, 2025b).** Claude-3.7-Sonnet's successor, balancing performance and speed, with low latency and high resource efficiency, excels in code generation.

**Grok-2-Vision(xAI, 2024).** The multi-modal model of xAI combines language and visual processing capabilities to handle various images and documents, and supports multilingual recognition and style analysis.

**Qwen-VL-Max.** The closed-source flagship model of Alibaba's Qwen series has been optimized for deployment in enterprise-level multimodal tasks, supporting joint input of images, text, videos, and others, with ultra-large parameter volume and high inference capability.

**Gemini-2.5-Pro(Gemini Team, 2025).**A multimodal model from Google DeepMind that achieves state-of-the-art performance on frontier reasoning and coding benchmarks. It excels at multimodal

understanding, including the ability to process up to 3 hours of video content and convert it into interactive code. Its combination of long context, multimodality, and enhanced reasoning capabilities unlocks new agentic workflows and complex problem-solving.

# G  ERROR CASES STUDY

## G.1  SIGNAL LEVEL

We observe that the model struggles to distinguish localized noise from clean signals in the spectrum quality assessment task. For example, given the question "Does this spectrum show obvious signal quality issues?", the ground-truth label was "Localized noise" or "Very low noise, eligible", indicating minor but noticeable signal interference. However, the model incorrectly predicted "No, the signal is very clear", resulting in a failed case. This misclassification reveals a key limitation: the model tends to overestimate the clarity of the spectrum when the noise is not global or strongly pronounced. In visual inspection, localized artifacts—though subtle—can be clearly identified by human annotators, whereas the model often dismisses them as negligible. It lacks sufficient sensitivity to weak or local signal distortions, or has overfit to globally noisy or clean examples during training, causing it to ignore partial imperfections. This insight aligns with our general observation: the model often fails to distinguish noise from true signal, especially when the noise is spatially sparse or located at the margins of the image. Such behavior may stem from the fact that the model treats the entire spectrum as a holistic input, and lacks mechanisms to perform fine-grained regional quality assessment. Additionally, for models not inherently multi-modal, spectra are often encoded as image representations and then passed through vision encoders or captioning modules, potentially discarding low-level noise patterns. As a result, noise may not be retained in the model's internal representation, leading to overly optimistic predictions.

## G.2  PERCEPTION LEVEL

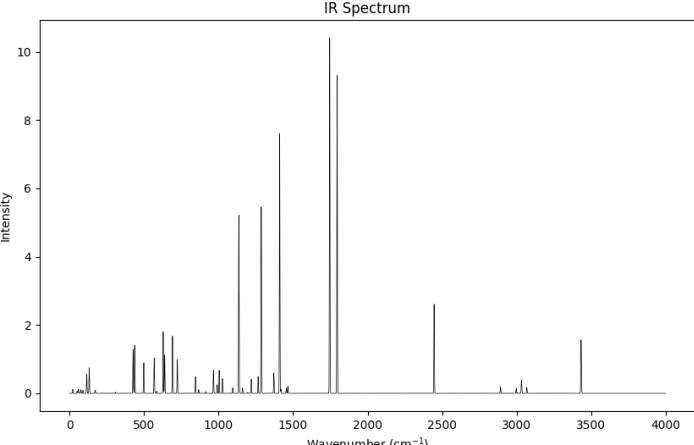

Figure 19: A Case of Functional Group Recognition

We found that for functional group recognition and peak assignment tasks, large language models such as Doubao-1.5-pro-thinking often fail to produce chemically accurate predictions, even when the visual features in the spectra are clear to human experts. For instance, in the functional group recognition task (Figure 19), the infrared (IR) spectrum exhibits a strong absorption band characteristic of a **carbonyl group (C=O)**, typically near 1700 cm$^{-1}$. However, the model incorrectly predicted **hydroxyl group (-OH)**. This suggests that the model likely over-relied on the presence of a broad peak or baseline shift, possibly mistaking low-intensity or overlapping signals for OH-stretching vibrations. In the peak assignment task (Figure 20), given the molecular formula $C_{10}H_7Cl$ and a clear singlet near 6.8 ppm in the $^1$H-NMR spectrum, the expected answer was **aromatic CH**

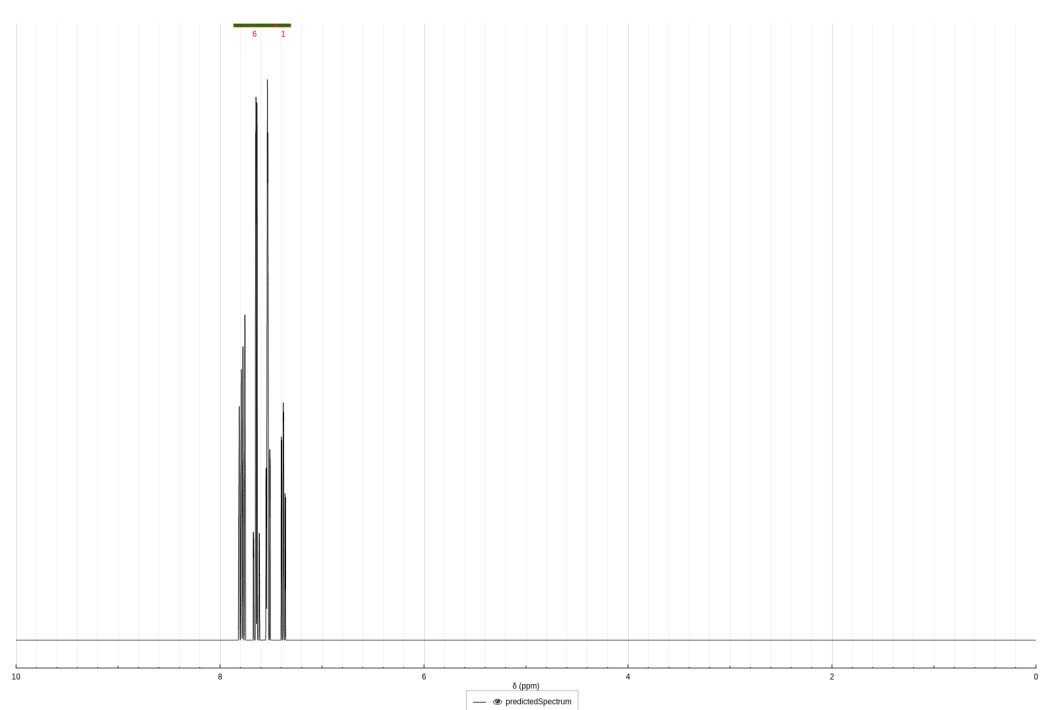

Figure 20: A Case of Peak Assignment

**next to a double bond**, i.e., a non-substituted position in the naphthalene ring. Yet the model responded with **aromatic CH adjacent to Cl**, a chemically invalid assignment considering the splitting pattern and electronic environment. This indicates a lack of fine-grained chemical reasoning and possibly an overemphasis on token-level keyword association rather than structural context. These cases expose the model's semantic-level misunderstanding, which goes beyond visual misinterpretation and highlights a deficiency in chemically grounded reasoning. We hypothesize two contributing factors. Firstly, the model may rely heavily on language priors, rather than truly integrating spectral visual features with molecular structure. Secondly, it lacks domain-specific supervision. Pretraining on generic data may not sufficiently expose the model to physical rules of spectroscopy, such as electron-withdrawing effects, chemical shift theory, or group frequency ranges.

### G.3 SEMANTIC LEVEL

At the semantic level, tasks involving **molecular structure elucidation** and **multi-modal reasoning** remain particularly challenging. Consider the example below:

In this case, the model is asked: *"The molecular formula of the compound is $C_4H_8O_2$. Use this information together with the provided IR spectrum image to infer possible structural features."* The correct answer should be **Ether**, based on the absence of a strong carbonyl absorption near 1700 cm$^{-1}$ and the elemental composition. However, the model incorrectly predicts **Carboxylic acid**, likely due to over-reliance on superficial signal patterns that resemble O–H stretching or C=O bands.

Even when the molecular formula is omitted (pure spectrum-based reasoning), the model continues to produce incorrect predictions, revealing a deficiency in cross-modal semantic alignment. This suggests that while LLMs may perform well on shallow text-image associations, they struggle with integrating spectral data and chemical constraints in a chemically meaningful way.

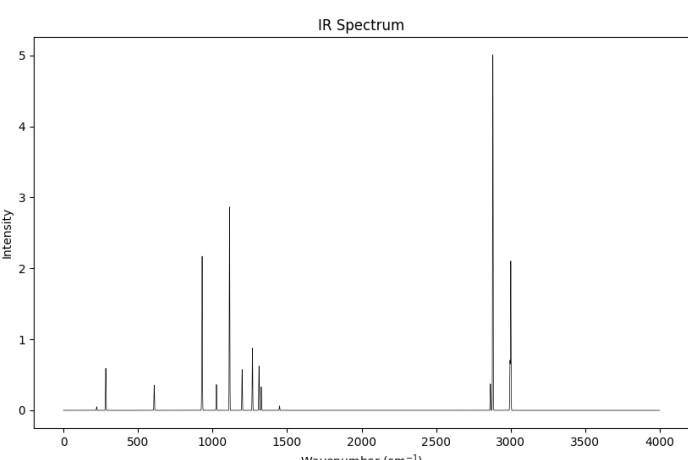

Figure 21: A Case of Fusing Multi-Modalities

### G.4 GENERATION LEVEL

Not surprisingly, the performance on generation tasks—especially structure generation—is significantly worse. This suggests that while models like **Claude-3.7-Sonnet** perform well on earlier levels such as perception, syntactic understanding, and basic semantic reasoning, they still struggle with more complex **forward problems** that require inferring new molecular structures from spectral data. **De novo generation** and **inverse problems** (e.g., predicting spectra from structure) pose even greater challenges, as they demand deeper chemical understanding and cross-modal generalization. In these settings, most models exhibit clear signs of overfitting or default to high-frequency patterns seen in training data.

Surprisingly, **Doubao-1.5-Vision-Pro-Thinking** demonstrates promising performance on forward problems, aligning well with its strong results in earlier semantic-level tasks such as functional group recognition, peak assignment, and molecular structure elucidation. This consistency suggests that the model may have a better internal representation of cross-modal chemical semantics, though its capability still falls short in full generation settings.

## H  MODEL ACCURACY VS. TOKEN ASSUMPTIONS

We conduct a comparative analysis of several Multimodal Large Language Models (MLLMs) from both semantic and generative levels, focusing on three representative tasks: Molecule Elucidation (ME), Fusing Spectroscopic Modalities (FM), and Forward Problems (FP). As shown in Figure 22, the performance gap among models is significant. Notably, models with lower average token assumptions, such as *DeepSeek-VL2*, tend to exhibit lower accuracy. In contrast, models with higher token assumptions, such as *Doubao-1.5-Vision-Pro-Thinking*, achieve superior performance, especially on complex *de novo* generation tasks like FP. This suggests that a longer reasoning chain, reflected in higher token usage, benefits complex problem-solving. However, the trade-off is increased computational cost and significantly longer inference time. These results highlight the efficiency-performance dilemma in MLLMs.

## I  DETAILED DATA STRUCTURE

This section details the comprehensive seed datasets curation pipeline and the three primary data structures that underpin our framework: the foundational **seed datasets**, the structured **benchmark data**, and the standardized **evaluation results**.

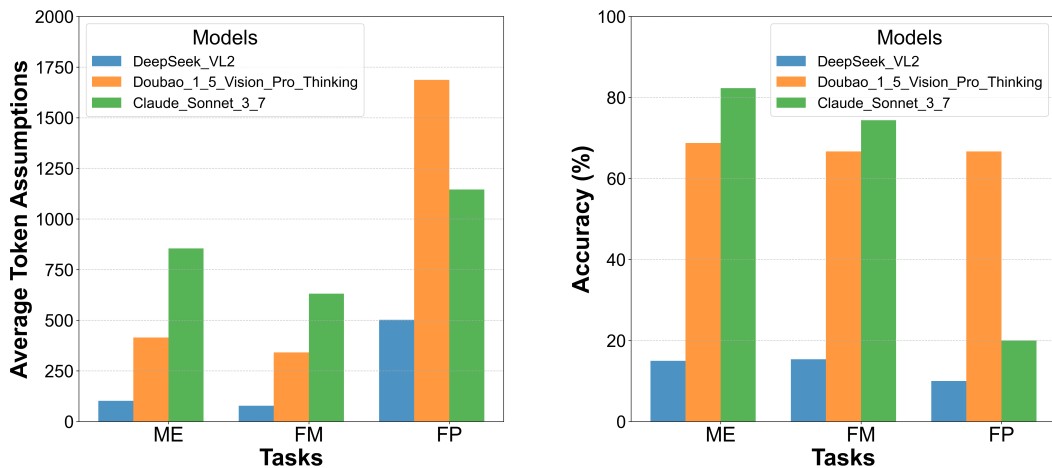

Figure 22: Model accuracy aligns with the model size.

## I.1 SEED DATA CURATION DETAILS

The seed datasets are curated from three primary sources to ensure both diversity and scientific rigor:

1. **Proprietary collections and in-house experimental data**: These include unpublished spectroscopic measurements and curated datasets from our collaborating laboratories. This source comprises approximately 238,869 molecular data points covering 8 types of spectra, offering higher authenticity and usability compared to most computationally generated spectra.

2. **Public repositories and benchmark datasets**: We integrate data from a range of widely recognized and authoritative sources, including SDBS (of Advanced Industrial Science & , AIST), QM9S (Zou et al., 2023b), NovoBench (Zhou et al., 2024a), and MolPuzzle (Guo et al., 2024b), among others. In total, seven distinct repositories and public datasets are used, collectively encompassing over 1.01 million unique chemical compounds.

3. **Literature mining**: Spectral data are systematically extracted from the *Supporting Information* sections of peer-reviewed publications, with a focus on articles from leading journals such as the *Journal of the American Chemical Society* (**JACS**) and *ACS Catalysis*.

All collected datasets undergo a unified processing pipeline that systematically maps each entry into three core chemical spaces: SMILES, molecular formula, and spectra. The resulting seed datasets are organized at the level of individual chemical substances, with each record containing the compound's SMILES, molecular formula, and a structured set of associated spectra, all stored in a standardized JSON format. This robust foundation facilitates downstream annotation and interoperability.

## I.2 SEED DATASETS STRUCTURE

The seed dataset is constructed by extracting essential information from raw experimental data, serving as the foundation for benchmark generation. Each entry contains a molecular index, SMILES string, molecular formula, and a list of associated spectra. An illustrative structure is provided in Listing 1. The `path` field is a list that may contain multiple files for a given spectrum type, accommodating cases such as multiple mass spectra for a single molecule.

Listing 1: Example structure of a seed dataset entry.

```
{
  "molecule_index": "MOL_0001",
  "smiles": "CCCCC1=CC=CC=C1",
  "formula": "C10H14",
  "spectra": [
```

```
        {"spectrum_type": "IR", "path": ["IR/MOL_0001.png"]},
        {"spectrum_type": "MASS", "path": ["MASS/MOL_0001.jpg",
        "MASS/MOL_0001_2.jpg"]},
        {"spectrum_type": "C-NMR", "path": ["C-NMR/MOL_0001.png"]},
        {"spectrum_type": "H-NMR", "path": ["H-NMR/MOL_0001.png"]}
    ]
}
```

## I.3 SPECTRUMBENCH DATA STRUCTURE

The benchmark data structure is designed to support a diverse range of tasks, including signal interpretation, perception, and semantic understanding. Each entry includes a unique identifier, image path(s), question, answer choices, ground truth answer, category, sub-category, data source, and timestamp. A representative example is shown in Listing 2. After processing by Spectrum-Lab, three additional fields are appended: `model_response` (the model's reasoning and output), `model_prediction` (the answer extracted from the model response), and `pass` (a boolean indicating whether the model's prediction matches the ground truth).

Listing 2: Example of a benchmark data entry.

```
{
  "id": "Perception_a9cf_250723_235951_318294",
  "image_path": [
    "data/Perception/Basic Property Prediction/Perception_a9cf_q.png"
  ],
  "question": "Given the mass spectrum image, what is the most likely
  molecular ion peak (m/z) observed for this compound?",
  "choices": ["85", "90", "120", "133"],
  "answer": "133",
  "category": "Perception",
  "sub_category": "Basic Property Prediction",
  "source": "",
  "timestamp": "2025-07-23 23:59:51"
}
```

## I.4 EVALUATION RESULTS STRUCTURE

The evaluation results structure records the model's predictions and performance for each benchmark instance. Listing 3 illustrates the format. For all data structures, the `image_path` field is specified relative to the `data` directory to ensure clarity and reproducibility. This standardized design facilitates systematic benchmarking and transparent evaluation across a wide range of spectroscopic machine learning tasks.

Listing 3: Example of an evaluation results entry.

```
{
  "id": "Signal_9131_250723_110552_245529_2",
  "image_path": [
    "data/Signal/Spectrum Type Classification/Signal_9131_2_q.png"
  ],
  "question": "What type of spectrum is shown in the image?",
  "choices": [
    "Infrared Spectrum (IR)",
    "Proton Nuclear Magnetic Resonance (H-NMR)",
    "Mass Spectrometry (MS)",
    "Carbon Nuclear Magnetic Resonance (C-NMR)"
  ],
  "answer": "Mass Spectrometry (MS)",
  "category": "Signal",
  "sub_category": "Spectrum Type Classification",
  "source": "",
  "timestamp": "2025-07-23 11:05:52",
  "model_prediction": "Mass Spectrometry (MS)",
```

```
    "model_response": "\\answer{Mass Spectrometry (MS)}",
    "pass": true
}
```

## J  COST ANALYSIS

To ensure consistency and fairness across all experiments, SpectrumLab employs a unified model interface and conducts all inference via API services, regardless of whether the underlying models are open-source or proprietary. This standardized evaluation pipeline enables direct and equitable comparison of model performance. With the exception of the generation-level scoring model, each benchmark run requires an average of 572 model invocations. The use of remote APIs introduces network latency, resulting in variability in inference times. Depending on the model architecture and complexity, the total time required to complete the full SpectrumBench benchmark ranges from approximately 40 minutes to 2 hours. For each model, we systematically record the overall inference time and the estimated monetary cost associated with completing the benchmark.

Given the current benchmark prompts and SpectrumLab's prompt engineering design, a complete run of the benchmark requires approximately 1,219,083 input tokens and 41,522 output tokens (as measured on InternVL3-78B, this figure is provided for reference only). Models with more elaborate reasoning or "thinking" capabilities may incur even higher token consumption.

Table 10 summarizes the key statistics for representative models evaluated in this study.

Table 10: Resource consumption and cost for representative models on the full SpectrumBench benchmark.

| Model | Inference Time (min) | Cost (USD) |
|---|---|---|
| Claude-3.5-Haiku | 99 | $0.94 |
| Claude-3.5-Sonnet | 70 | $7.47 |
| Claude-4-Opus | 123 | $24.00 |
| Claude-4-Sonnet | 90 | $11.66 |
| GPT-4o | 103 | $4.23 |
| GPT-4-Vision-Preview | 113 | $8.08 |
| GPT-4.1-2025-04-14 | 103 | $1.54 |
| Grok-2-Vision | 62 | $2.12 |
| InternVL3-78B | 120 | N/A |

## K  USAGE OF LARGE LANGUAGE MODELS IN THIS MANUSCRIPT

In preparing this manuscript, we used a large language model (LLM) solely for editorial purposes. Its functions were limited to proofreading for typographical errors, correcting grammatical mistakes, and enhancing the clarity and readability of the text.

## L  LIMITATIONS

While this work introduces the concept of SpectrumWorld , it is important to acknowledge that the field of AI for Spectroscopy remains in its nascent stages, we recognize several limitations within our primary contributions, SpectrumBench and SpectrumLab .

**Limitations of SpectrumBench** First, regarding Task Format, SpectrumBench currently supports only multiple-choice and a limited number of open-ended questions. While this design is suitable for Large Language Models (LLMs), it is insufficient for evaluating a broader range of machine learning models, such as Convolutional Neural Networks (CNNs) and Graph Neural Networks (GNNs), as discussed in our introduction. Second, concerning Spectrum Type, although we have incorporated a wide array of spectrum types compared to previous works (Lu et al., 2025; Xu et al., 2025; Bushuiev et al., 2024b; Zhou et al., 2024a), several crucial spectroscopic modalities remain uncovered. Notable examples include X-ray Diffraction (XRD) (Guo et al., 2024a; Salgado et al., 2023)

and fluorescence spectra (Parker & Rees, 1962), which are vital for comprehensive material characterization. Finally, addressing Spectroscopic Task Type, spectroscopy techniques are fundamental across diverse scientific disciplines, including physics, astronomy, chemistry, and biology, primarily for characterizing substances like molecules, proteins, peptides, and SMILES sequences. From the perspective of LLMs, a generic categorization of modalities into "text" and "images" is inadequate for representing the complexity of data. The inherent diversity of spectroscopic modalities complicates the immediate definition of all possible tasks. Consequently, SpectrumBench presently lacks important benchmarks in several areas, such as spectrum-spectrum retrieval (Curry et al., 1969; Wang et al., 2022; Lu et al., 2025) and peptide sequence analysis (Zhou et al., 2024a). We acknowledge that it will be challenging for SpectrumBench to encompass all relevant tasks in the near future, and we aim to foster collaborative efforts with the community and various laboratories to collectively advance the development of AI in spectroscopy.

**Limitations of SpectrumLab** Our second main contribution, SpectrumLab, also presents certain limitations. Firstly, regarding its data functionality, while SpectrumLab successfully unifies seed datasets and provides data curation tools-SpectrumAnnotator, it currently lacks tools for the preprocessing and segmentation of raw data across multiple spectroscopic modalities. Secondly, concerning metrics, the current evaluation framework within SpectrumLab is relatively simplistic, relying primarily on accuracy and a lenient, LLM-based scoring method for open-ended questions. In future iterations, we plan to define and incorporate a broader array of task-specific metrics to enable more nuanced and robust model evaluation.

