# OpenReview forum: "SpectrumWorld: Artificial Intelligence Foundation for Spectroscopy"
_ICLR.cc/2026/Conference — Submitted to ICLR 2026_

### Official Review · Reviewer_fyXy · 2025-10-22

**Soundness:** 2
**Presentation:** 2
**Contribution:** 2
**Rating:** 4
**Confidence:** 4

**Summary:**

The paper introduces SpectrumLab, an LLM-centric platform for creating benchmarks across spectroscopic and spectrometric data modalities and for evaluating LLMs on these benchmarks. The authors test a wide range of state-of-the-art LLMs and release a well-documented codebase.

**Strengths:**

- Identifying molecules from spectroscopic measurements is crucial across many scientific domains, yet existing benchmarks are limited. This work proposes an extensible approach to benchmarking in these fields.
- The paper unifies datasets from multiple spectrometric modalities into a single resource, including some in-house, previously unpublished data.
- A new taxonomy for spectrometric annotation tasks is proposed.

**Weaknesses:**

- Although the paper proposes a benchmark, it contains little exploratory data analysis (EDA) of the seed datasets. For instance, Appendix G.1 does not list all data sources—a fundamental requirement for any new benchmark. Without EDA, it is difficult to assess the benchmark’s quality and significance. By EDA I mean, for example, what molecules are contained in the dataset, how noisy the spectra are, what instruments were used to generate the spectra, etc.
- The emphasis on multimodality is insufficiently justified:
  - The text states, “As illustrated in Figure 1, this comprehensive data foundation accurately reflects the diverse and complex multi-modal spectroscopic scenarios encountered in real-world applications.” Please provide a concrete example where multiple spectroscopy types must be annotated simultaneously. In natural products research, for example, high-throughput LC-MS/MS is often followed by NMR of selected isolated molecules; these steps are typically separate, and the overall pipeline would likely benefit more from stronger single-modal models for LC-MS/MS and for NMR than from a multimodal approach.
  - Strong multimodal models usually require overlapping training examples across modalities (e.g., different spectra for the same molecules). Can the authors demonstrate that the aggregated dataset offers meaningful intersections between modalities? Without such overlap, the aggregation is hard to justify.
- The platform is heavily LLM-driven and as a result many LLM-generated tasks do not reflect practical needs. Several tasks look like like textbook exercises where machine learning is unnecessary, while others are so difficult that general-purpose LLMs are unlikely to help. For example, “Spectrum Type Classification” or “Basic Feature Extraction” seem of limited real-world relevance; lines 986–994 describe a task that appears solvable without ML; and the task on lines 976–986 is unclear even for a human. Conversely, “Molecular Structure Elucidation” is extremely challenging even for domain- and modality-specific SOTA models.
- The manuscript contains numerous typos (e.g., lines 367–368 or end of line 298).

**Questions:**

- Can the authors better demonstrate the practical utility of the proposed resource? How would spectroscopic machine learning community and practitioners benefit from the proposed platform?

---

> ### Author Response · Authors · 2025-11-19
> **Reply to Reviewer fyXy - Part 1/2**
>
> We are grateful for the detailed comments. Here, we present our responses corresponding to the questions you've asked.
>
> **Q1: Lack of Exploratory Data Analysis**
>
> A1: Thank you for raising this important point. We fully agree that adequate EDA is essential for assessing dataset quality. In the revised version, we have added a dedicated expanded EDA section(**Appendix C.1**) that reports. These additions provide the transparency required for evaluating the benchmark’s scientific value and data reliability.
>
> **Q2: Justification for a Multimodal Benchmark**
>
> A2: As now explained in **Appendix C.2**(Multimodal Justification), our motivation for constructing a multimodal benchmark comes from two observations:
>
> 1.Scientific complementarity of spectroscopic modalities.
>
> LC–MS/MS provides mass and fragmentation patterns, whereas NMR offers fine-grained substructure clues; SMILES/graphs contribute symbolic structural priors. Many recent studies in molecular analysis and  demonstrate the value of combining such heterogeneous signals. For example, recent studies show that adding formula constraints and multiple spectrum types significantly improves structure elucidation accuracy [1][2]. Multimodality is not a stylistic choice but reflects how spectroscopy is practiced in real scientific settings.
>
> 2.Natural intersections exist in real datasets.
>
> Contrary to forced aggregation, our EDA (**Appendix C.1**) shows that **a substantial portion of molecules naturally contain paired modalities** (e.g., MS + NMR + SMILES). We have added a table quantifying these cross-modal overlaps.
>
> Table: Multimodal completeness and modality pairing across datasets in SpectrumWorld. “Fully
> paired” indicates that each molecule contains all listed modalities.
>
>  | Dataset                         | Modalities Included                                  | Fully Paired? | % Molecules with Complete Set |
> |---------------------------------|-------------------------------------------------------|--------------|-------------------------------|
> | QM9S                            | UV, IR, Raman                                         | Yes          | 100%                          |
> | Multimodal-Spectroscopic-Datasets | C-NMR, H-NMR, HSQC-NMR, MS⁺, MS⁻, IR               | Yes          | 100%                          |
> | MolPuzzle                       | IR, MS, HNMR, CNMR                                    | Yes          | 100%                          |
> | SDBS                            | IR, MS, HNMR, CNMR, Raman, ESR                        | No           | Variable (2–91%)              |
>
> Table: Distribution of spectral modalities within the SDBS dataset. Each row reports the number
> of spectra and the percentage of molecules containing that modality
>
> | Modality | #Spectra | % Molecules Containing This Modality |
> |----------|----------|---------------------------------------|
> | IR       | 39,980   | 91.64%                                |
> | MS       | 32,838   | 75.27%                                |
> | HNMR     | 18,317   | 41.98%                                |
> | CNMR     | 17,688   | 40.54%                                |
> | Raman    | 4,569    | 10.47%                                |
> | ESR      | 924      | 2.12%                                 |
>
> Taken together, these observations justify our multimodal design: it mirrors real-world spectroscopy workflows and leverages the complementary and co-occurring signal structure present in existing datasets. We emphasize that all experimental datasets used in SpectrumBench are publicly available or will be made public upon acceptance, enabling full transparency and reproducibility.
>
> ---
>
> [1] Yao, L. et al. (2023). Conditional Molecular Generation Net Enables Automated Structure Elucidation Based on ¹³C NMR Spectra and Prior Knowledge. https://doi.org/10.1021/acs.analchem.2c05817
>
> [2] Jonas, E. (2019). Deep imitation learning for molecular inverse problems. https://proceedings.neurips.cc/paper_files/paper/2019/file/b0bef4c9a6e50d43880191492d4fc827-Paper.pdf

---

> ### Author Response · Authors · 2025-11-19
> **Reply to Reviewer fyXy - Part 2/2**
>
> **Q3: LLM-generated tasks do not reflect actual needs? ("too easy" vs. "too hard")**
>
> A3: We acknowledge this concern and would like to clarify the role of SpectrumBench:
>
> - The purpose is capability probing, not domain replacement.
>   - Even elementary perception tasks (e.g., spectrum quality assessment, generation tasks) reveal substantial weaknesses in current MLLMs. These tasks are intentionally simple so that ML/NLP researchers can enter spectroscopy AI without requiring domain-specific training.
> - Difficult tasks intentionally expose spectrum-specific reasoning gaps.
>   - Challenging tasks such as “Molecular Structure Elucidation” are not meant to be solved perfectly today. Instead, they provide a stress test revealing the sharp drop in MLLM performance when moving from perception → semantic understanding → chemical reasoning → generation. This layer-wise degradation is a core insight provided by SpectrumBench.
>
> In this sense, the mixture of “easy” and “hard” tasks is not a flaw but a necessary design choice: it yields a structured, multi-level evaluation that highlights where MLLMs succeed, where they fail, and where domain-specific methods are still indispensable.
>
> **Q4: Practical Utility for the Spectroscopy ML Community**
>
> A4: Thank you for this important question. SpectrumWorld was designed from the outset to provide concrete, practical benefits to both ML researchers and spectroscopy practitioners. It offers: (i) a unified and standardized task taxonomy that resolves long-standing fragmentation in the spectroscopy-ML literature; (ii) a fully reproducible curation pipeline (SpectrumAnnotator and SpectrumVerifier) for constructing dataset variants, cleaning spectra, and aligning multimodal inputs; (iii) a cross-modality evaluation interface that mirrors real analytical workflows where MS, NMR, IR, and structural representations jointly inform decision-making; and (iv) an extensible benchmark that already supports 20+ MLLMs and will be expanded with domain-specific models and metrics in upcoming releases. In practice, these components provide a common framework that both communities can adopt, reuse, and build upon with minimal overhead.
>
> **Q5: Typos and Minor Issues**
>
> A5: All typographical issues identified by the reviewer have been corrected in the revised manuscript.
>
> **Closing Statement**
>
> We thank the reviewer for the high-level perspective and domain-specific insights. Due to space limitations in the comment box, several clarifications and additional results are provided in the corresponding appendices, and we kindly invite you to refer to them for fuller context. We hope our responses have addressed your concerns, and we warmly welcome any further suggestions. Several of your points overlap with feedback from other reviewers, and the corresponding responses may offer additional clarification.

---

> ### Author Response · Authors · 2025-11-24
> **Looking forward to your reply**
>
> Dear Reviewer,
>
> Thank you for your insightful comments and suggestions. We have submitted response to address the points you’ve raised, including clarifications and revisions to improve the presentation and completeness of the work. We hope that these will sufficiently address your concerns and lead to an improved rating of our work. Should you have any further questions or require additional clarification, please do not hesitate to ask.
>
> We look forward to your updated review and remain open to further discussions. Thank you again for your time and consideration.

---

> ### Author Response · Authors · 2025-11-28
> **Hoping to continue the discussion**
>
> Dear Reviewer,
>
> We appreciate your participation in this productive discussion. We hope all your doubts have been addressed. If so, we humbly request that you consider increasing the score.
>
> We also encourage you to review our responses to other reviewers who raised important points and are satisfied with our explanations and additional experiments. Thank you!

---

### Official Review · Reviewer_we1U · 2025-10-25

**Soundness:** 3
**Presentation:** 2
**Contribution:** 2
**Rating:** 4
**Confidence:** 4

**Summary:**

The authors introduce a "platform" for AI research in spectroscopy. The platform has multiple compoments, including Python libraries and a benchmark. The paper benchmarks many recent multimodal LLMs on the tasks.

**Strengths:**

The benchmark library and other Python libraries introduced here seem useful for a specific community.

**Weaknesses:**

The manuscript is extremely dense, very hard to read, also due to the introduction of more than 20 abbreviations, and the use of a lot of domain jargon. The relationship between different components (SpectrumBench, SpectumLab) is hard to understand.

The modalities that are actually being used are often unclear. Figure 1 mainly talks about images and text, while other parts discuss model inputs and outputs without specifying the modality or data representation (e.g. molecules, spectra, ...).

The SpectrumLab introduced here seems to be only addressing multimodal language models, and not other, more task-specific models. It seems like a lot of the work of the past years regarding (single- or multi-modal) representation learning of spectra and molecular structures seems to be incompatible with this work, and certainly not benchmarked here.

Overall, the benefit for the general ML community seems limited. The paper is mainly speaking to a very specific, application-oriented community focusing on spectral data. If the authors' target audience are computer scientists, then parts of the manuscript should be rephrased and improved, to make it more generally understandbale, introduce or remove community jargon, and make sure a more mathematically correct and consistent nomenclature regarding data types is used, and potentially also a more abstract task specification should be introduced.

**Questions:**

Figure 1: Why is the word "image" used so often? The modality is not an image if you have smiles codes or spectra. Smiles codes are strings that can also be represented as graphs. Spectra are lists of peaks or lists of intensities, so essentially vectors. Neither of them are images, but of course they can be visualized in images. Or do you call those image modalities, because you are actually using them as images for the multi-modal foundation models in your benchmark?

Figure 2 introduces many architectures, such as GNN, CNN, MLPs. Why are those discarded in your benchmark? Which of the tasks in your benchmark could also be solved without the text modality, or with a hybrid model that does not depend on an image representation of molecules and spectra?

How is the "Text-to-Any (De novo Generation)" task related to spectra? How is it different from what the entire community of generative models for molecules and materials is doing?

What does GLM4.5V in Figure 5 mean?
How can there be a unique ground truth output in the bottom right box in Figure 5? This is a generation task that can only sample but not predict "correct" answers.

Minor: Most of the fonts in Figure 1 are too small. Please improve Figure 1 for readability.

---

> ### Author Response · Authors · 2025-11-19
> **Reply to Reviewer we1U - Part 1/2**
>
> We are grateful for the detailed comments. Here, we present our responses corresponding to the questions you've asked.
>
> **Q1: On abbreviations and domain-specific jargon**
>
> A1: We admit that the initial submission introduced a large number of abbreviations and domain-specific terms, which may pose a barrier for ML readers. As discussed in the paper, a long-standing challenge in spectroscopy AI is the absence of a unified nomenclature—prior works often use inconsistent task names and heterogeneous input–output definitions. One of the core contributions of SpectrumBench is organizing these fragmented tasks into a coherent structure.
> To make this organization clearer, we have added an updated figure in **Appendix B**.
>
> Figure 8 summarizes how we derive the SpectrumBench taxonomy. Starting from general spectra problems(left), we group tasks by input-output type into four coarse families: forward problems (molecule → spectrum), inverse problems (spectrum → molecule), de novo generation (text → any molecular/spectral / peptide sequence), and understanding (any → text, e.g., classification or explanation). Each family is then instantiated as the 14 concrete sub-tasks used in SpectrumBench (middle blocks), which are further abstracted into the four semantic levels on the right—Signal → Perception → Semantic → Generation—used throughout the paper.
>
> We call molecule → spectrum a **forward problem** because the molecular structure encodes the full physico-chemical state, while the measured spectrum is a compressed observation of that state; predicting spectra from a known structure is therefore the forward direction of the underlying physical model. Conversely, spectrum → molecule is an **inverse problem**, where one attempts to recover a richer latent description (the molecule) from partial, noisy observations (the spectra).
>
> We will revise the manuscript to explain this hierarchy and terminology more explicitly, ensuring that the taxonomy is fully accessible to ML readers.
>
> **Q2: SpectrumLab appears tailored only to MLLMs rather than domain-specific models**
>
> A2: Thank you for pointing this out. We agree that domain-specific spectroscopic models are important baselines. Our choice to focus the first release on MLLMs is methodological: MLLMs are currently the only class of models that can be evaluated uniformly across all 14 tasks and 10+ modalities using a single text–image interface, without introducing incompatible preprocessing or task-specific engineering. In contrast, existing spectral and molecular models rely on heterogeneous formats and pipelines, making fair comparison impossible at this stage. **This choice reflects a comparability consideration for the current version, rather than a constraint of the overall framework.** Domain models are fully within the scope of SpectrumWorld, and our roadmap explicitly includes adding domain-specific baselines and metrics(see **Appendix L** for details).
>
> **Q3: Why is the word “image” used for SMILES or spectra?**
> A3: We appreciate this careful observation. We fully agree with the reviewer: in their scientific form, spectra are lists of peaks or intensity vectors. These are **not images in the scientific sense**. To unify evaluation across MLLMs, we visualize spectral signals as images solely for input compatibility with vision encoders. This follows the standard multimodal LLM research[1, 2].  We have clarified this distinction and removed any ambiguous labeling in Figure 1. There are two different meanings of “modality” relevant here:
> 1. Domain-level modalities (scientific meaning):
>   - SMILES (string)
>   - Molecular graphs
>   - ¹H-NMR and ¹³C-NMR (vector/intensity list)
>   - MS/IR (peak-intensity modality)
> 2. Model-level modalities (MLLM meaning):
>   - Text input
>   - Image input (the expected interface for visual encoders)
>
> To avoid confusion in the revised manuscript, we will state:
> **“We do not equate SMILES or spectra with images in the scientific sense; ‘image’ refers only to the representation consumed by the MLLM’s visual encoder.”**
>
> Again, thank you for the attentive reading—your comment significantly improves clarity.
>
> ---
>
> [1] Li, J. et al. (2023). BLIP-2: Bootstrapping Language-Image Pre-training with Frozen Image Encoders and Large Language Models.
>
> [2] Chen, X. et al. (2023). PaLI-X: On Scaling up a Multilingual Vision and Language Model.

---

> ### Author Response · Authors · 2025-11-19
> **Reply to Reviewer we1U - Part 2/2**
>
> **Q4: How is the “Text-to-Any”(De novo Generation) task related to spectroscopy?**
>
> A4: Thanks for the question. With the updated Figure 8, we hope the terminology and taxonomy are now clearer. The Text-to-Any task models the way spectroscopists often formulate problems in natural language, for example, requesting molecules consistent with partial spectral clues or describing expected spectral properties. In this sense, the task reflects a realistic entry point in spectroscopy workflows, where textual hypotheses are mapped to structural, spectral, or molecular outputs.”
>
> **Q5: Figure 5 – GLM4.5V and the “unique ground truth”**
>
> A5: Thank you for pointing this out. For generation tasks, the appropriate term is ‘reference answer’ rather than ‘ground truth’.   We have updated the caption accordingly in the revised manuscript.  In Figure 5, “GLM4.5V” simply indicates that the molecule shown was generated by the GLM-4.5V model.
>
> **Closing Remark**
>
> We appreciate the reviewer’s detailed comments on presentation, terminology, and task definitions—these suggestions have been genuinely helpful in improving the clarity of the manuscript. We hope our responses adequately address your concerns. Due to limitations in the comment box, several clarifications and additional results are provided in the corresponding appendices, and we kindly invite you to refer to them for fuller context.  Should you have further suggestions, we warmly welcome them. Several of the raised points intersect with feedback from other reviewers, and the corresponding responses may offer additional clarification.

---

> ### Author Response · Authors · 2025-11-24
> **Looking forward to your reply**
>
> Dear Reviewer,
>
> Thank you for your insightful comments and suggestions. We have submitted response to address the points you’ve raised, including clarifications and revisions to improve the presentation and completeness of the work. We hope that these will sufficiently address your concerns and lead to an improved rating of our work. Should you have any further questions or require additional clarification, please do not hesitate to ask.
>
> We look forward to your updated review and remain open to further discussions. Thank you again for your time and consideration.

---

> > ### Comment · Reviewer_we1U · 2025-11-25
> >
> > Thank you for answering my questions. While these partially clarify the manuscript, my main conclusion still remains true: The benefit for the general ML community seems overall limited. The task definitions, modalities, data representations, etc. seem all extremely specific to spectroscopy, limiting accessibility for the general ML community to enter this field.
> > It still remains unclear to me what is actually contained in the data. If you say that you are using images, is the raw spectral data (i.e. a table) contained in your benchmark or only the image? And if you want to introduce an actual benchmark, what exactly is the task? Your paper sounds like the task is to take images as input (e.g. in MLLMs). Your answer sounds like any data representation of the data is allowed. The second case would be more interesting, but at the same time, the manuscripts lack alternative models beyond MLLMs to solve the tasks.
> > Furthermore, there seems to be a general misunderstanding of how to benchmark generative tasks. Counting the errors with respect to "ground truth" or "reference answer" is not appropriate, no matter how it is called.
> >
> > In light of this, I thank the authors for answering my questions, but I do not adjust my score. I believe a lot more work would be needed, and I remain unsure if an ML conference has the right audience for this manuscript.

---

> > > ### Author Response · Authors · 2025-11-26
> > > **Reply to Reviewer we1U**
> > >
> > > Thank you for the follow-up comments. We understand and respect your current evaluation, and we appreciate the opportunity to clarify the remaining points. Several aspects of your response suggest there may still be misunderstandings about the scope and design objectives of the benchmark. Below, we provide concise clarifications corresponding to each concern:
> > >
> > > ---
> > >
> > > **Q1: On whether task definitions, modalities, and data representations are overly spectroscopy-specific**
> > >
> > > A1: SpectrumBench is grounded in spectroscopy applications, but the evaluation interface is not domain-specific. Regardless of their scientific origin, all modalities in the benchmark are ultimately standardized into **text and image inputs**, consistent with common multimodal evaluation settings such as **VQA-RAD** [1] and **PathVQA**[2], where highly domain-specific medical data are also formatted as text–image pairs for general ML models to use.
> > >
> > > We also provide unified APIs for evaluation, annotation tools (SpectrumAnnotator) for constructing SFT and RL training data, and downloadable processed seed datasets (**currently 100+ downloads by external users**), so models can be evaluated **without requiring any spectroscopy background**. If there are specific parts that still feel inaccessible, we would appreciate concrete feedback so that we can address them in the next revision.
> > >
> > > [1] Lau, J. J., Gayen, S., Ben Abacha, A., & Demner-Fushman, D. (2018).  A dataset of clinically generated visual questions and answers about radiology images. Proceedings of the EMNLP Workshop on Machine Translation.
> > >
> > > [2] He, L., Zhou, X., Huang, L., & Wang, X. (2020). PathVQA:  Pathology Visual Question Answering. Proceedings of IEEE CVPR Workshops.
> > >
> > > **Q2: On whether raw data are included or only images**
> > >
> > > A2: As shown in Table 6, except for MolPuzzle (which is inherently image-based), all other spectral datasets retain their original numeric sequence form (peak lists or intensity vectors). During benchmark construction, these raw sequences were first visualized into image form to create the seed datasets, and subsequently annotated to form the final benchmark. This workflow is illustrated in Figure 3.
> > >
> > > **Q3: On the absence of non-MLLM baselines**
> > >
> > > A3: SpectrumBench v1 evaluates only MLLMs. The rationale is methodological rather than conceptual:
> > > - Existing baselines for NMR, IR, Raman, and MS use heterogeneous pipelines and incompatible input formats, making fair comparison difficult without significant standardization.
> > > - The goal of this first version is to answer a specific question:
> > > - “How well can general-purpose MLLMs perform when exposed to spectroscopy tasks under a unified interface?”
> > >
> > > Based on reviewer suggestions, the revised manuscript will adopt a more precise wording (e.g., SpectrumVQA) to indicate this scope.
> > >
> > > **Q4: On generative task evaluation**
> > >
> > > A4: We do not evaluate generative tasks via accuracy or token-level error rates. Instead, following recent work on LLM-as-a-judge evaluation paradigms for open-ended generation[3][4], we use an LLM-as-judge setup to assess whether the generated outputs contain plausible reasoning chains and satisfy expected chemical or spectral constraints. Appendix A.1 reports robustness checks for this evaluation protocol, and the resulting scores are consistent with performance patterns observed at the other three evaluation levels.
> > >
> > > [3] Zheng, L., Chiang, W.-L., Sheng, Y., et al. (2023). Judging LLM-as-a-Judge with MT-Bench and Chatbot Arena. arXiv:2306.05685.
> > >
> > > [4] Liu, Y., Liu, P., & Cohan, A. (2025). On Evaluating LLM Alignment by Evaluating LLMs as Judges. NeurIPS.
> > >
> > > ---
> > >
> > > **We appreciate the time invested in reviewing this work and your continued engagement. Regardless of whether the score is updated, your feedback has been valuable in strengthening and refining the work.**

---

### Official Review · Reviewer_paW5 · 2025-10-29

**Soundness:** 3
**Presentation:** 3
**Contribution:** 3
**Rating:** 6
**Confidence:** 3

**Summary:**

This paper presents **SpectrumWorld**, an AI infrastructure designed for spectroscopy, consisting of two main components: the **SpectrumLab** toolkit/platform and the large-scale benchmark **SpectrumBench**. SpectrumBench organizes evaluations along a four-level task taxonomy—**Signal → Perception → Semantic → Generation**—and provides a unified definition of tasks and evaluation protocols across multiple modalities (spectra, molecular graphs/SMILES, and natural language). Its goal is to systematically assess the capabilities of multimodal large language models (MLLMs) in spectroscopy.

The authors further design an automated benchmark generation and annotation module, **SpectrumAnnotator**, complemented by **SpectrumVerifier** and human-in-the-loop validation to establish a closed-loop quality control pipeline. The entire framework covers data collection, cleaning and standardization, automated and manual annotation, evaluation, and leaderboard publication. Experiments benchmark over 20 mainstream MLLMs under a unified setting, reporting both overall strengths and weaknesses, along with a detailed analysis of resource usage and computational cost.

**Strengths:**

- Clear infrastructural positioning and broad coverage:

The system systematically abstracts spectroscopy tasks into a four-level hierarchy and provides a modular evaluation ecosystem (tasks, evaluators, leaderboards), offering an extensible common platform for the research community.

- Reproducible data and annotation workflow:

The proposed SpectrumAnnotator automatically constructs multimodal QA/generation samples, and together with SpectrumVerifier and expert review forms a closed-loop quality control process that is transparent and reproducible.

- Extensive model evaluation with unified protocol:

Low- and mid-level tasks are unified as four-choice questions, while generation tasks are scored by GPT-4o, producing a single accuracy metric for easy comparison across 23 open- and closed-source MLLMs.

- Well-motivated problem statement:

The paper clearly identifies key pain points in spectroscopy AI—such as the domain gap between experimental and theoretical spectra, modality heterogeneity, and the lack of standardized benchmarks—and designs SpectrumWorld to address them.

- Openness and community potential:

Anonymous release of code, datasets, and data schemas encourages long-term community extension and reuse.

**Weaknesses:**

- Lack of quantitative evidence for annotation quality:

Although the paper describes the quality control process, it lacks systematic statistics on annotation quality (e.g., inter-annotator agreement, error rate, or comparison between automatic and manual annotations). Providing such quantitative analysis or visualization would strengthen the reliability of the benchmark.

- Single-dimensional evaluation metrics:

For generative tasks, using GPT-4o as the sole evaluator may introduce evaluation bias. In addition, the benchmark mainly relies on accuracy as the metric, without sufficient task-specific measures.

**Questions:**

- Evaluation bias and robustness:

For generative tasks evaluated using GPT-4o, could there be systematic bias favoring certain model families? Have you considered using multiple evaluators or calibration methods (e.g., open-source LLM judges or rule-based metrics) to verify the consistency of evaluation results?

- Task-specific metrics:

Beyond overall accuracy, do you plan to introduce domain-specific metrics for tasks such as de novo peptide sequencing (peptide precision) ?

---

> ### Author Response · Authors · 2025-11-19
> **Reply to Reviewer paW5 - Part 1/2**
>
> Thanks for your insightful comments. We will respond to each of your points in the following.
>
> **Q1: Lack of quantitative evidence for annotation quality.**
>
> A1: We appreciate this important point. In fact, we would like to share this experience. SpectrumBench employs a mixed automatic–human curation pipeline, and our early iterations indeed contained non-trivial issues—including (i) mismatches between question descriptions and the associated spectra or structures, (ii) incorrect or ambiguous reference answers, and (iii) unintended overlaps across task types. These issues arose partly from model limitations (e.g., OCR and insufficient domain knowledge) and partly from under-constrained prompt templates.
>
> Through multiple cycles of prompt revision and human-in-the-loop review, annotation reliability improved substantially. To address the reviewer's request for quantitative evidence, we have added a new figure and a summary table in **Appendix E**, reporting (1) a direct comparison between automatic and manual annotations across all task types, and (2) the reduction in error rates before and after prompt refinements. The reported error rates refer only to the raw outputs of the SpectrumAnnotator; the final benchmark includes an additional manual verification stage and therefore does not contain these errors. We hope these additions clearly document the effectiveness of the iterative curation process and resolve the concerns regarding annotation quality.
>
> **Q2: Generation level, Single-dimensional evaluation metrics validation?**
>
> A2: We agree that using GPT-4o as the sole evaluator may introduce bias, and we appreciate the reviewer raising this concern. Our intention in this version was to probe how much general spectroscopic knowledge current MLLMs already possess, not to position LLM-based evaluators as definitive ground truth.
>
> To address this concern, we have constructed a second version of the generation-task prompt, shifting from “predict structure directly” to “provide a meaningful reasoning trace with explicit scoring criteria”. This reduces variability associated with single-dimensional similarity metrics and mitigates evaluator bias.
>
> In addition, we conducted a cross-evaluator validation using three independent evaluators: GPT-4o, Intern-S1, and Claude-Opus-4. All three evaluators produced highly consistent trends. These results indicate that our main conclusion—that current MLLMs still struggle with spectroscopic generation and reasoning—does not depend on a specific evaluator. The updated prompt, additional results, and a correlation analysis across evaluators are included in **Appendix A.1**.

---

> ### Author Response · Authors · 2025-11-19
> **Reply to Reviewer paW5 - Part 2/2**
>
> **Q3: Lack of domain-specific metrics? Do you plan to introduce domain-specific metrics for tasks such as de novo peptide sequencing(peptide precision)?**
>
> A3: We fully agree that. We plan to integrate domain-specific metrics, and your suggestion (e.g., peptide-level precision for de novo sequencing) is an excellent example. This direction is fully aligned with our roadmap and is explicitly listed as a limitation in **Appendix L**.
>
> Our guiding principle is to maintain consistency and extensibility in the benchmark. To avoid mixing incompatible pipelines across modalities, we first need to identify the key spectroscopic tasks, survey the corresponding literature, and determine the commonly accepted domain metrics before adding them into SpectrumLab in a systematic way.
> To demonstrate progress, we have already surveyed the next wave of domain-specific metrics we plan to integrate. Examples include:
> - Structure Elucidation: Canonical SMILES match, Graph isomorphism(molecule graph), atom count match[1, 2, 3]
> - Peptide sequence design: Amino acid-level and peptide-level precision/recall, Coverage, Accuracy, Error Rate, AUC,  Post-Translation Modification(PTM) [4, 5, 6, 7]
> - Spectral Library Matching: Top-N Accuracy, Cosine Similarity, Dot Product, Pearson Correlation Coefficient, MAE [8, 9]
>
> If the reviewer has additional suggestions or domain-preferred metrics, we would be very grateful to incorporate them!
>
> **Closing Remark.**
>
> We sincerely appreciate your constructive feedback and hope our responses have addressed your concerns. Due to limitations in the comment box, several clarifications and additional results are provided in the corresponding appendices, and we kindly invite you to refer to them for fuller context.  If you have further suggestions, we warmly welcome them. Some of your points overlap with comments from other reviewers, and the corresponding responses may offer additional clarification.
>
> ---
>
> [1] Alberts, M. et al. (2024). Leveraging infrared spectroscopy for automated structure elucidation. https://doi.org/10.1038/s42004-024-01341-w
>
> [2] Yilmaz, M. et al. (2022). De novo mass spectrometry peptide sequencing with a transformer model. https://proceedings.mlr.press/v162/yilmaz22a.html
>
> [3] Alberts, M. et al. (2023). Learning the Language of NMR: Structure Elucidation using Transformer Models. https://doi.org/10.26434/chemrxiv-2023-8wxcz
>
> [4] Zhou, J. et al. (2024). NovoBench: Benchmarking Deep Learning-based De Novo Peptide Sequencing. https://arxiv.org/abs/2406.11906
>
> [5] Yang, T. et al. (2024). π-HelixNovo for practical large-scale de novo peptide sequencing. https://doi.org/10.1093/bib/bbae021
>
> [6] Zhang, J. et al. (2012). PEAKS DB: de novo sequencing assisted database search. https://doi.org/10.1074/mcp.M111.010587
>
> [7] Petrovskiy, D. et al. (2024). PowerNovo: de novo peptide sequencing using transformers. https://doi.org/10.1038/s41598-024-65861-0
>
> [8] Kim, S. & Zhang, X. (2013). Comparative analysis of mass spectral similarity measures. https://doi.org/10.1155/2013/509761
>
> [9] Huber, F. et al. (2021). Spec2Vec: Improved mass spectral similarity scoring. https://doi.org/10.1371/journal.pcbi.1008724

---

> ### Author Response · Authors · 2025-11-24
> **Looking forward to your reply**
>
> Dear Reviewer,
>
> Thank you for your insightful comments and suggestions. We have submitted response to address the points you’ve raised, including clarifications and revisions to improve the presentation and completeness of the work. We hope that these will sufficiently address your concerns and enhance the clarity and completeness of our work. Should you have any further questions or require additional clarification, please do not hesitate to ask.
>
> We look forward to your updated review and remain open to further discussions. Thank you again for your time and consideration.

---

> > ### Comment · Reviewer_paW5 · 2025-11-24
> >
> > Thanks for the author's response. I maintain my positive score.

---

### Official Review · Reviewer_trcg · 2025-11-01

**Soundness:** 2
**Presentation:** 2
**Contribution:** 2
**Rating:** 4
**Confidence:** 3

**Summary:**

This paper introduces SpectrumLab, a modular toolkit and leaderboard for AI in spectroscopy, and SpectrumBench, a hierarchical benchmark spanning four levels (signal, perception, semantic, generation), 14 tasks, and 10+ spectrum types. The dataset is built from seed data encompassing >1.2M distinct chemical compounds, with a mixed auto+human curation pipeline. The authors evaluate 23 MLLMs and report large performance gaps between low-level recognition and higher-level reasoning/generation. Code and evaluation pipelines are packaged for reproducibility.

**Strengths:**

* The benchmark is hierarchically structured to mirror real spectroscopy workflows (signal → perception → semantic → generation) across roughly 14 task types and more than 10 modalities, which makes failure modes and capability gaps observable.
* Coverage of 20+ multimodal LLMs under a single evaluation protocol yields stable empirical patterns, strong performance on low-level recognition but weak generation/reasoning.
* The curation pipeline combines automated item generation with human-in-the-loop verification for better quality assurance.

**Weaknesses:**

* The scope is limited to MLLMs, with no baselines from domain-specific models (e.g., neural network trained on molecular strings and spectra data). Including such baselines and perhaps a hybrid setting where an LLM orchestrates specialized models for reasoning would yield a more informative comparison.
* Due to the focus on MLLMs, the task design and metrics are narrow. Reliance on multiple choice and plain accuracy underutilizes domain-specific measures (e.g., spectral similarity, shift MAE).
* Open-ended generation is graded by an LLM judge (GPT-4o), enabling scale but risking bias and score inconsistency, and may lead to metric validity concerns. How can we be sure of the quality of LLM judge-based evaluations?

Minor:
* Presentation can be improved. For example, the texts in Figure 1 and Figure 3 are very hard to read when printed on papers.

**Questions:**

Please refer to weaknesses.

---

> ### Author Response · Authors · 2025-11-19
> **Reply to Reviewer trcg - Part 1/3**
>
> We appreciate your thoughtful questions and suggestions. Below, we have organized your queries along with our responses for clarity:
>
> **Q1: “Why only compare against MLLMs? Why not include specialized domain models or hybrid systems?”**
>
> A1: Thanks for raising this important point. We fully agree that domain-specific models are valuable baselines. In this version, we focus on MLLMs for three methodological reasons.
>
> **(1)Heterogeneous pipelines prevent fair comparison at this stage.**  As discussed in the paper,  domain models across NMR, IR, Raman, and MS rely on incompatible preprocessing, representations, and training protocols. Without substantial standardization, comparisons would be confounded by differences in pipeline engineering rather than model capability.
>
> **(2)Our primary objective of SpectrumBench(V1) is to answer a first-order scientific question:**
>
> “How far can today’s general-purpose MLLMs go in spectroscopy?” MLLMs uniquely support all tasks through a text/image interface, enabling consistent and modality-agnostic benchmarking.
>
> **(3)Domain models and MLLMs are complementary, not competing. SpectrumWorld will support both.**
>
> Our long-term vision is exactly aligned with reviewer’suggestion: SpectrumWorld will incorporate domain-specific baselines, domain-specific metrics. However, including heterogeneous domain models before identifying crucial tasks and establishing evaluation pipelines would not yield fair or meaningful comparisons.
>
> **Q2: “Why not design a layered system that incorporates specialized domain models, with MLLMs coordinating them?”**
>
> A2: This is an excellent suggestion and aligns closely with the direction outlined in our roadmap. As noted in Q1, our long-term vision is not for MLLMs to replace domain-specific spectroscopic models, but rather to coordinate them across the full laboratory workflow—covering signal preprocessing, perception, semantic understanding and reasoning-based generation.
>
> A hybrid architecture is indeed a promising paradigm:
> - Domain models handle modality-specific expertise (e.g., a special kind of spectrum analysis)
> - MLLMs integrate these results, perform reasoning, and interact with human chemists
>
> We view this as a realistic and powerful path toward AI-assisted spectroscopy. However, implementing such a system in the current benchmark is limited by the issues raised in Q1, making hybrid baselines difficult to include fairly at this stage and would lead to a system too complex for users to easily replicate or use.

---

> ### Author Response · Authors · 2025-11-19
> **Reply to Reviewer trcg - Part 2/3**
>
> **Q3: “Metrics rely mostly on accuracy and do not include domain-specific measures.”**
>
> A3: We agree that domain-specific metrics are essential for fully evaluating spectroscopic performance. Our choice of accuracy-based metrics in V1 is due to a design constraint: to maintain comparability across more than 14 tasks and 10+ modalities under a unified multimodal setting. Meanwhile, we have already released and open-sourced the full AI-ready spectroscopic datasets, which have received many community downloads and usage, and we are now building the necessary modality-specific preprocessing pipelines to support more specialized metrics.
>
> Our design principle for incorporating domain models and domain-specific metrics is to proceed systematically: identify the core tasks studied in spectroscopy, survey the benchmarks and datasets used in prior literature, and integrate the corresponding evaluation toolkits—including preprocessing and metric implementations—into SpectrumLab. Once these pipelines mature, the MLLM-compatible infrastructure we provide will naturally extend to domain-specific baselines as well. This process requires careful engineering and validation, which is why the current version uses accuracy-based metrics.
>
> As stated in the paper’s Limitations section(**See Appendix L**), SpectrumLab will include broader task-specific metrics in future iterations, and all related tools are already open-sourced to encourage community contributions.
>
> To demonstrate progress, we have already surveyed the next wave of domain-specific metrics we plan to integrate. Examples include:
> - Structure Elucidation: Canonical SMILES match, Graph isomorphism(molecule graph), atom count match[1, 2, 3]
> - Peptide sequence design: Amino acid-level and peptide-level precision/recall, Coverage, Accuracy, Error Rate, AUC,  Post-Translation Modification(PTM) [4, 5, 6, 7]
> - Spectral Library Matching: Top-N Accuracy, Cosine Similarity, Dot Product, Pearson Correlation Coefficient, MAE [8, 9]
>
> If the reviewer has additional suggestions or domain-preferred metrics, we would be very grateful to incorporate them!
>
> **Q4:” Concerns about using LLMs as evaluators in generation tasks.”**
>
> A4: We agree that using LLMs as evaluators may introduce bias, and we appreciate the reviewer highlighting this issue. Our intention in the generation tasks is not to claim precise correctness, but to examine how much general spectroscopic knowledge current MLLMs possess and whether they can produce meaningful reasoning trajectories. The V1 benchmark indeed had limitations—not because the tasks were inappropriate, but because the scoring rubric placed disproportionate weight on output modality and relied on broad, loosely defined similarity criteria. In response, we redesigned a second version of the evaluation prompt with a more rigorous and structured rubric that emphasizes scientifically meaningful reasoning and provides clearly defined score bands.
>
> To further validate robustness, we applied the updated prompts to multiple independent evaluators (GPT4o, Intern-S1, and Claude Opus 4). All evaluators produced consistent trends, reinforcing that our main conclusion—MLLMs still struggle with generation and reasoning—does not depend on a particular choice of evaluator. The revised prompts, additional results, and a correlation matrix and scatter analysis demonstrating cross-evaluator consistency are provided in **Appendix A.1**.
>
>
> ---
>
> [1] Alberts, M. et al. (2024). Leveraging infrared spectroscopy for automated structure elucidation. https://doi.org/10.1038/s42004-024-01341-w
>
> [2] Yilmaz, M. et al. (2022). De novo mass spectrometry peptide sequencing with a transformer model. https://proceedings.mlr.press/v162/yilmaz22a.html
>
> [3] Alberts, M. et al. (2023). Learning the Language of NMR: Structure Elucidation using Transformer Models. https://doi.org/10.26434/chemrxiv-2023-8wxcz
>
> [4] Zhou, J. et al. (2024). NovoBench: Benchmarking Deep Learning-based De Novo Peptide Sequencing. https://arxiv.org/abs/2406.11906
>
> [5] Yang, T. et al. (2024). π-HelixNovo for practical large-scale de novo peptide sequencing. https://doi.org/10.1093/bib/bbae021
>
> [6] Zhang, J. et al. (2012). PEAKS DB: de novo sequencing assisted database search. https://doi.org/10.1074/mcp.M111.010587
>
> [7] Petrovskiy, D. et al. (2024). PowerNovo: de novo peptide sequencing using transformers. https://doi.org/10.1038/s41598-024-65861-0
>
> [8] Kim, S. & Zhang, X. (2013). Comparative analysis of mass spectral similarity measures. https://doi.org/10.1155/2013/509761
>
> [9] Huber, F. et al. (2021). Spec2Vec: Improved mass spectral similarity scoring. https://doi.org/10.1371/journal.pcbi.1008724

---

> ### Author Response · Authors · 2025-11-19
> **Reply to Reviewer trcg - Part 3/3**
>
> Table: Correlation Matrix between Evaluator:
> | Evaluator | gpt-4o | gpt-4o-again | intern-s1 | claude-opus-4 |
> |-----------|--------|--------------|-----------|---------------|
> | gpt-4o | 1.000 | 0.962 | 0.934 | 0.933 |
> | gpt-4o-again | 0.962 | 1.000 | 0.919 | 0.953 |
> | intern-s1 | 0.934 | 0.919 | 1.000 | 0.935 |
> | claude-opus-4 | 0.933 | 0.953 | 0.935 | 1.000 |
>
> All pairwise correlations between evaluators exceed **0.91**, indicating exceptionally strong inter-rater agreement. In particular, the correlation between **GPT-4o and GPT-4o-again reaches 0.962**, further confirming the stability and repeatability of the evaluation process.
>
> ---
>
> Table: Score Variance for Each Model
> | Model | Mean | Std | Variance |
> |-------|------|-----|----------|
> | CLAUDE_OPUS_4 | 0.593 | 0.035 | 0.001 |
> | LLAMA_VISION_90B | 0.547 | 0.069 | 0.005 |
> | Grok_2_Vision | 0.482 | 0.066 | 0.004 |
> | Doubao_1_5_Vision_Pro | 0.476 | 0.049 | 0.002 |
> | Qwen_2_5_VL_72B | 0.436 | 0.059 | 0.004 |
> | Qwen_VL_Max | 0.416 | 0.087 | 0.008 |
> | Qwen_2_5_VL_32B | 0.385 | 0.058 | 0.003 |
> | intern-s1-notink | 0.344 | 0.060 | 0.004 |
> | intern-s1-mini | 0.341 | 0.038 | 0.001 |
> | intern-s1-think | 0.328 | 0.083 | 0.007 |
> | DeepSeek_VL2 | 0.153 | 0.097 | 0.009 |
>
> Regarding the model-wise statistics, the table summarizes the mean, standard deviation, and variance of scores assigned by four evaluators across 11 models. Overall, the results show high consistency across evaluators: most models exhibit very small standard deviations (9 out of 11 are ≤ 0.069), and all variances remain below 0.01. Even the models with the largest dispersion still show only modest variation relative to their mean scores. These patterns collectively indicate that evaluator judgments are tightly clustered, with minimal disagreement across models.
>
> **Closing Remark.**
>
> We sincerely appreciate your constructive feedback and hope our responses have addressed your concerns. Due to limitations in the comment box, several clarifications and additional results are provided in the corresponding appendices, and we kindly invite you to refer to them for fuller context. If you have further suggestions, we warmly welcome them. Some of your points overlap with comments from other reviewers, and the corresponding responses may offer additional clarification.

---

> ### Author Response · Authors · 2025-11-24
> **Looking forward to your reply**
>
> Dear Reviewer,
>
> Thank you for your insightful comments and suggestions. We have submitted response to address the points you’ve raised, including clarifications and additional experimental results. We hope that these will sufficiently address your concerns and lead to an improved rating of our work. Should you have any further questions or require additional clarification, please do not hesitate to ask.
>
> We look forward to your updated review and remain open to further discussions. Thanks so much.

---

> ### Author Response · Authors · 2025-11-28
> **Hoping to continue the discussion**
>
> Dear Reviewer,
>
>
> We appreciate your participation in this productive discussion. We hope all your doubts have been addressed. If so, we humbly request that you consider increasing the score.
>
> We also encourage you to review our responses to other reviewers who raised important points and are satisfied with our explanations and additional experiments. Thank you!

---

### Author Response · Authors · 2025-12-01
**Thank All Reviewers**

To all reviewers,

We would like to sincerely thank all reviewers for their thoughtful insights and valuable comments. These discussions have significantly helped us improve the quality and clarity of our manuscript.

It seems that the vacancy of a summarization of the contributions and the positioning of this benchmark have caused some misunderstandings regarding its relevance to the ML community. We summarize our core contributions here and have highlighted them in the revised paper:

1. **First MLLM-Centric Spectroscopy Foundation**: We introduce **SpectrumLab**, the first comprehensive infrastructure designed to standardize heterogeneous spectroscopic tasks (14 tasks, 10+ modalities) into a unified interface. This addresses the long-standing fragmentation in scientific data, making spectroscopy accessible to general ML researchers without requiring domain-specific preprocessing expertise;
2. **A "Stress Test" for MLLM Reasoning**: We construct **SpectrumBench**, a hierarchical benchmark that reveals a critical insight: while current SOTA MLLMs (e.g., GPT-4o, Claude) perform well on low-level perception, they exhibit a sharp performance drop in high-level semantic reasoning and generation. This "layer-wise degradation" provides the ML community with a challenging testbed for evaluating complex reasoning and cross-modal alignment capabilities beyond standard VQA tasks;
3. **High-Quality Data Pipeline**: We propose **SpectrumAnnotator**, a mixed automatic-human curation pipeline. We have quantitatively verified its reliability, reducing annotation error rates significantly through iterative refinement (e.g., from ~50% to ~20% in raw outputs, followed by manual verification);
4. **Reproducibility & Openness**: We release the full toolkit, code, and datasets (which have already received 100+ downloads), enabling the community to easily replicate results and extend the benchmark to new models.

We are excited that you recognized our contributions. We quote correspondingly as below:

- **On Infrastructure & Novelty**: "The benchmark is hierarchically structured to mirror real spectroscopy workflows... which makes failure modes and capability gaps observable." [Reviewer trcg]; "The system systematically abstracts spectroscopy tasks... providing an extensible common platform." [Reviewer paW5]; "Identifying molecules from spectroscopic measurements is crucial... This work proposes an extensible approach." [Reviewer fyXy].
- **On Coverage & Scale**: "Coverage of 20+ multimodal LLMs under a single evaluation protocol yields stable empirical patterns." [Reviewer trcg]; "Clear infrastructural positioning and broad coverage." [Reviewer paW5].
- **On Toolkit Utility**: "The curation pipeline combines automated item generation with human-in-the-loop verification for better quality assurance." [Reviewer trcg]; "release a well-documented codebase." [Reviewer fyXy].

We also appreciate the constructive suggestions, based on which we have significantly improved our manuscript. The main changes are:

- **Robust Evaluation (Addressing Bias)**: We expanded the evaluation protocol for generation tasks. Instead of relying solely on GPT-4o, we added **cross-evaluator validation using Claude-3-Opus and Intern-S1**, demonstrating high consistency (correlation > 0.9) and proving that our conclusions are not evaluator-biased [Appendix A.1].
- **Data Quality Assurance**: We added a detailed analysis of **annotation error rates** (before and after prompt refinement) and a comparison between automatic and manual annotations to quantitatively prove data reliability [Appendix E].
- **Comprehensive EDA**: We added a dedicated **Exploratory Data Analysis (EDA)** section to visualize data distribution and spectral properties [Appendix C.1].
- **Clarified Terminology**: We revised the manuscript to explicitly distinguish between "scientific images" and "visual representations for MLLMs," and added a **Task Hierarchy Figure** [Appendix B] to make the taxonomy accessible to general ML readers.
- **Baselines & Roadmap**: We clarified the methodological choice of focusing on MLLMs for this version (to ensure unified comparability) while explicitly outlining the roadmap for integrating domain-specific baselines and metrics [Appendix L].

We would again like to thank all reviewers for their time and effort. We hope that our extensive new experiments and clarifications adequately address all concerns.

Sincerely,

Authors

---

### Note · Authors · 2025-10-30

I have read and agree with the venue's withdrawal policy on behalf of myself and my co-authors.

---

> ### Note · Program_Chairs · 2025-10-30
>
> We approve the reversion of withdrawn submission.

---

### Meta-Review · Area_Chair_Gjhy · 2026-01-08

**Summary:**

The submission introduces SpectrumLab, a unified platform intended to standardize and accelerate deep learning research in spectroscopy. Reviewers acknowledge the novel contribution of a comprehensive benchmark platform for spectroscopy AI, the broad coverage and a clear hierarchical organization of tasks, and a open design to allow human in the loop and modular toolkit incorporation.

Reviewers also expressed concerns including:
1. The exclusive focus on multimodal LLMs and the absence of domain-specific or task-specialized baselines may hinder the professional nature of the platform.
2. Major evaluation metrics rely on a single LLM judge, whose reliability and validity may not be sufficiently serious. Domain-specific or physics-informed measures are expected.
3. Annotation quality evidence (e.g., agreement statistics, error rates) is lacking, making it hard to assess benchmark reliability.
4. The presentation appears dense with intensive jargons for general ML audience.

**Reviewer Concerns:**

From the authors' rebuttal, the following updates/conclusions are drawn.
1. The authors made a reasonable explanation of only compare against MLLMs, but it still seems less relevant to acknowledged practice. Even in the heterogeneous multi-modal case, one could still develop a domain-specific model that could be more dedicated and professional to the tasks.
2. I understand the author described situation and status, but I still perceive the lack of domain-specific metrics and evaluators make the platform less professional and less connected to the science community. The consensus among LLMs is not a direct demonstration of the validity of evaluation results (e.g., they may have similar hallucination patterns, e.g., over smoothing).
3. In the rebuttal the authors provided some examples of data quality check and modifications that lead to refinements. This is a good evidence, but does not seem a systematic treatment.
4. The authors made some improvements in the presentation, which is a good sign. Nevertheless, the major body of language still requires a refinement.

In all, I appreciate the effort and devotion into this platform as well as substantial replies and additional effort in the rebuttal. I would perceive the work as delicately designed but may appear not properly aligned with AI+Science community status and expectations on the tasks.

**Reviewer Scores:**

I would not expect essential improvement of the score after the rebuttal.

---

### Decision · Program_Chairs · 2026-01-26

Reject